# Identification of the trail-following pheromone receptor in termites

Souleymane Diallo[1,2], Kateřina Kašparová[1,3], Josef Šulc[1], Jibin Johny[2], Jan Křivánek[1], Jana Nebesářová[4,5], David Sillam-Dussès[6], Pavlína Kyjaková[1], Jiří Vondrášek[1], Aleš Machara[1], Ondřej Lukšan[1]*, Ewald Grosse-Wilde[2], Robert Hanus[1]*

[1]Institute of Organic Chemistry and Biochemistry of the Czech Academy of Sciences, Prague, Czech Republic; [2]Czech University of Life Sciences, Prague, Czech Republic; [3]Department of Ecology, Faculty of Science, Charles University, Prague, Czech Republic; [4]Laboratory of Electron Microscopy, Faculty of Science, Charles University, Prague, Czech Republic; [5]Biology Centre of the Czech Academy of Sciences, České Budějovice, Czech Republic; [6]University Sorbonne Paris Nord, Villetaneuse, France

## eLife Assessment

This **important** work by Diallo et al. substantially advances our understanding of the chemosensory system of a non-hymenopteran eusocial insect by identifying the first olfactory receptor for the trail pheromone in termites. The evidence supporting the conclusions that the receptor PsimOR14 is very narrowly tuned for the pheromone neocembrene is **compelling**. The work will be of broad interest to entomologists, chemical ecologists, neuroscientists, and molecular biologists.

**\*For correspondence:**
ondrej.luksan@uochb.cas.cz (OL);
robert@uochb.cas.cz (RH)

**Competing interest:** The authors declare that no competing interests exist.

**Abstract** Pheromone communication is the cornerstone of eusocial insect societies since it mediates the social hierarchy, division of labor, and concerted activities of colony members. The current knowledge on molecular mechanisms of social insect pheromone detection by odorant receptors (ORs) is limited to bees and ants, while no OR was yet functionally characterized in termites, the oldest eusocial insect clade. Here, we present the first OR deorphanization in termites. We selected four OR sequences from the annotated antennal transcriptome of the termite *Prorhinotermes simplex* (Psammotermitidae), expressed them in Empty Neuron *Drosophila*, and functionally characterized them using single sensillum recording (SSR). For one of the selected ORs, PsimOR14, we obtained strong responses to the main component of *P. simplex* trail-following pheromone, the monocyclic diterpene neocembrene. PsimOR14 showed a narrow tuning to neocembrene with only one additional compound out of 67 tested generating non-negligible responses. We report on homology-based modeling and molecular dynamics simulations of ligand binding by PsimOR14. Subsequently, we used SSR in *P. simplex* workers and identified the olfactory sensillum responding to neocembrene, thus likely expressing *PsimOR14*. Finally, we demonstrate that *PsimOR14* is significantly more expressed in worker antennae compared to soldiers, which correlates with higher sensitivity of workers to neocembrene.

## Introduction

Chemical communication is the cornerstone of eusocial insect societies. It mediates the social hierarchy and division of labor, social cohesion, and concerted activities of colony members. Different clades of eusocial insects have independently evolved chemical signals acting in similar social contexts, such as trail, alarm and queen pheromones, colony and caste recognition cues, and other signals. The

convergent evolution of chemical signaling is best manifested by many similarities between ants and termites, in spite of the great evolutionary distance between these two dominant groups of social insects with great ecological significance (*Leonhardt et al., 2016*; *Tuma et al., 2020*).

Since the identification of the multigene family of insect odorant receptors (ORs) expressed in antennae and maxillary palps of *Drosophila melanogaster* (*Clyne et al., 1999*), neurophysiology of insect olfaction has seen great progress in terms of phylogenetic reconstructions of OR evolution across taxa, functional characterizations (deorphanizations) of multiple ORs and ultimately their recent structural characterizations (*Butterwick et al., 2018*; *Del Mármol et al., 2021*; *Wang et al., 2024*; *Zhao et al., 2024*). Even though the insect ORs have seven transmembrane domains like the ORs known from vertebrates, they do not share many tangible sequence similarities. Insect ORs have an inverted membrane topology in the dendrites of olfactory sensory neurons compared to vertebrate ORs (*Benton et al., 2006*; *Clyne et al., 1999*) and unlike the vertebrate ORs known to act as GPCR receptors, insect ORs function as odorant-gated ion channels (*Sato et al., 2008*; *Wicher et al., 2008*). While the ORs from the basal wingless insect lineage Archaeognatha are homotetramers (*Del Mármol et al., 2021*; *Brand et al., 2018*), ORs of all other Insecta form heteromeric complexes with a highly conserved coreceptor protein (ORCo) (*Brand et al., 2018*; *Butterwick et al., 2018*; *Larsson et al., 2004*; *Sato et al., 2008*). As recently shown in mosquitoes and aphids, these complexes consist of one OR and three ORCo subunits (*Wang et al., 2024*; *Zhao et al., 2024*).

The OR repertoire is greatly variable across Insecta and ranges from units in the wingless Archaeognatha and basal winged order Odonata to tens or hundreds ORs identified in most other flying insects (*Robertson, 2019*; *Yan et al., 2020*). Insect ORs often lack a clear orthology pattern across phylogeny, because the OR family evolved via rapid birth-and-death process, accompanied by multiple gene duplications, pseudogenizations, and losses. Lineage-specific expansions, together with a considerable variability in OR ligand specificities (from broad to narrow tuning), allow for a rapid response of olfactory system to ecological and life history changes (*Andersson et al., 2015*; *Benton, 2015*; *Nei and Rooney, 2005*; *Robertson, 2019*). Ecology-driven plasticity in OR evolution has been convincingly demonstrated by comparisons of specialist versus generalist insects, the former often having much lower repertoires of ORs and other chemosensory proteins (*Robertson, 2019*).

The knowledge on OR function and ligand specificities has historically been obtained mainly from deorphanization studies on *Drosophila* and other holometabolan insects using various heterologous expression systems. Recently, this bias was in part compensated by OR deorphanizations in more basal taxa, for example, the wingless Archaeognatha (*Del Mármol et al., 2021*), or the hemimetabolous aphids (*Zhang et al., 2017*; *Zhang et al., 2019a*) and locusts (*Guo et al., 2020*; *Chang et al., 2023*).

Within social insects, eusocial Hymenoptera received considerable attention both in terms of OR repertoire reconstructions and functional characterizations with multiple ORs being deorphanized in ants (*Pask et al., 2017*; *Slone et al., 2017*) and the honey bee (e.g., *Gomez Ramirez et al., 2023*; *Wanner et al., 2007*). The amassed knowledge suggests that the complex communication and orientation capabilities in the colonies of eusocial Hymenoptera are facilitated by the greatly expanded repertoire of ORs, especially that of the 9-exon subfamily in ants and paper wasps participating in the detection of cuticular hydrocarbons (CHCs) as important cues in contact chemoreception of colony and caste identity and fertility status in eusocial insects (*Engsontia et al., 2015*; *Legan et al., 2021*; *McKenzie et al., 2016*; *Pask et al., 2017*; *Zhou et al., 2015*). The 9-exon subfamily and the overall OR richness have been inherited by the extant eusocial Hymenoptera from the ancestor of Aculeata (*McKenzie et al., 2016*), and also solitary aculeate taxa display large OR repertoires *Obiero et al., 2021*; this preadaptation might have been important for the repeated emergence of eusociality and the related complex communication. This is supported by the reduction of OR array (including 9-exon genes) in parasitic ant taxa, along with the simplification of their behavioral repertoire (*Jongepier et al., 2022*).

The multiple convergences in biology and life histories between termites and ants call for comparison of olfactory detection of chemical signals and environmental cues in the two groups. Yet, despite relatively good knowledge on chemistry of termite pheromones and recognition cues (reviewed in *Bagnères and Hanus, 2015*; *Bordereau and Pasteels, 2011*; *Mitaka and Akino, 2021*), molecular aspects of olfaction remain largely understudied in termites. Termite ORs were so far only addressed with respect to their diversity, inferred from genome assemblies (*Harrison et al., 2018*; *Terrapon*

*et al., 2014*) or whole-body transcriptomes (*Mitaka et al., 2016*), recently complemented by comprehensive search for chemosensory protein repertoire using the antennal transcriptomes of three termite species (*Johny et al., 2023*). Termite ORs are organized in relatively conserved, highly orthologous pattern, and their total numbers range from 28 to 69 (*Johny et al., 2023*; *Terrapon et al., 2014*). These numbers are lower than in their solitary cockroach relative *Blattella germanica* (*Harrison et al., 2018*), and dramatically lower than in ants, having up to over 400 ORs (*Engsontia et al., 2015*; *Legan et al., 2021*; *McKenzie et al., 2016*; *Pask et al., 2017*; *Zhou et al., 2015*). Thus, termites clearly contradict the paradigm on eusociality as a driver of OR richness, even though their chemical communication is by far more complex than in solitary insects and includes pheromone components from a variety of chemical classes, such as fatty-acyl derived alcohols, aldehydes and ketones, terpenoids, and, last but not least, the CHCs. Independently of eusocial Hymenoptera, termites evolved an intricate communication system using CHCs as kin- and nestmate discrimination cues and as indicators of caste identity and fertility status (reviewed in *Bagnères and Hanus, 2015*; *Mitaka and Akino, 2021*). Therefore, compared to ants, the termite social evolution seemingly adopted a different trajectory to accommodate the needs of chemical communication, including CHC detection.

One of the alternative hypotheses springs from observations that termites possess an extraordinarily rich set of ionotropic receptors (IRs), reaching up to more than 100 in some species (*Harrison et al., 2018*; *Johny et al., 2023*; *Terrapon et al., 2014*). Even though the richness of IRs is shared by termites and their cockroach relatives, with *B. germanica* having the largest IR repertoire ever identified in insects (*Robertson et al., 2018*), IRs also underwent termite-specific expansions (*Harrison et al., 2018*; *Johny et al., 2023*). Since insect IRs have been shown to respond to volatile ligands (*Benton et al., 2009*), we cannot rule out that they also participate in the complex chemical communication in termites, as has been previously speculated (*Harrison et al., 2018*). Nevertheless, since no chemosensory proteins have yet been functionally characterized in termites, ORs remain the prime candidates for pheromone detection.

Here, we report on the first OR deorphanization in termites. We build on the knowledge about the chemical ecology of the Cuban subterranean termite *Prorhinotermes simplex* (Rhinotermitidae) (*Hanus et al., 2006*; *Hanus et al., 2009*; *Jirošová et al., 2017*; *Piskorski et al., 2007*), on the annotated repertoire of 50 ORs from *P. simplex* antennal transcriptome (*Johny et al., 2023*), on additional *P. simplex* sequencing data (caste-specific head transcriptomes, draft genome assembly), and on laboratory culture of the species. We select four *P. simplex* OR sequences, study their function by means of the Empty Neuron *Drosophila* expression system and single-sensillum recording (SSR) using panels of biologically relevant ligands including the *P. simplex* pheromones and chemically related compounds. We identify PsimOR14 as the pheromone receptor narrowly tuned to the monocyclic diterpene neocembrene, known as the major component of the trail-following pheromone (TFP) (*Sillam-Dussès et al., 2009*). We demonstrate a strong and selective response of PsimOR14 to neocembrene, with only one additional ligand, geranylgeraniol, generating non-negligible receptor response. We further report on homology-based modeling of neocembrene binding by PsimOR14 and perform molecular dynamics (MD) simulations to estimate the impact of ligand binding on PsimOR14 dynamicity. Finally, we identify the *P. simplex* olfactory sensillum specifically responding to neocembrene and document worker caste-biased expression of PsimOR14 accompanied by significantly higher sensitivity to neocembrene compared to *P. simplex* soldiers.

## Results
### Phylogenetic reconstruction and candidate OR selection

In the first step, we reconstructed the phylogeny of termite ORs and ORCos using published protein sequences from two species in combination with our antennal transcriptome data on three species and the bristletail *Lepisma saccharina* as basal insect outgroup. In the resulting tree, all ORCo sequences and ORs from *L. saccharina* were basally situated, while the majority of termite OR sequences were organized into two large sister clusters, both of which were further split into several sub-branches mostly containing one sequence from all five termite species (*Figure 1A*). Only a few exceptions to this highly orthologous pattern were spotted, such as isolated sequences or species-specific expansions with a maximum of four paralogs.

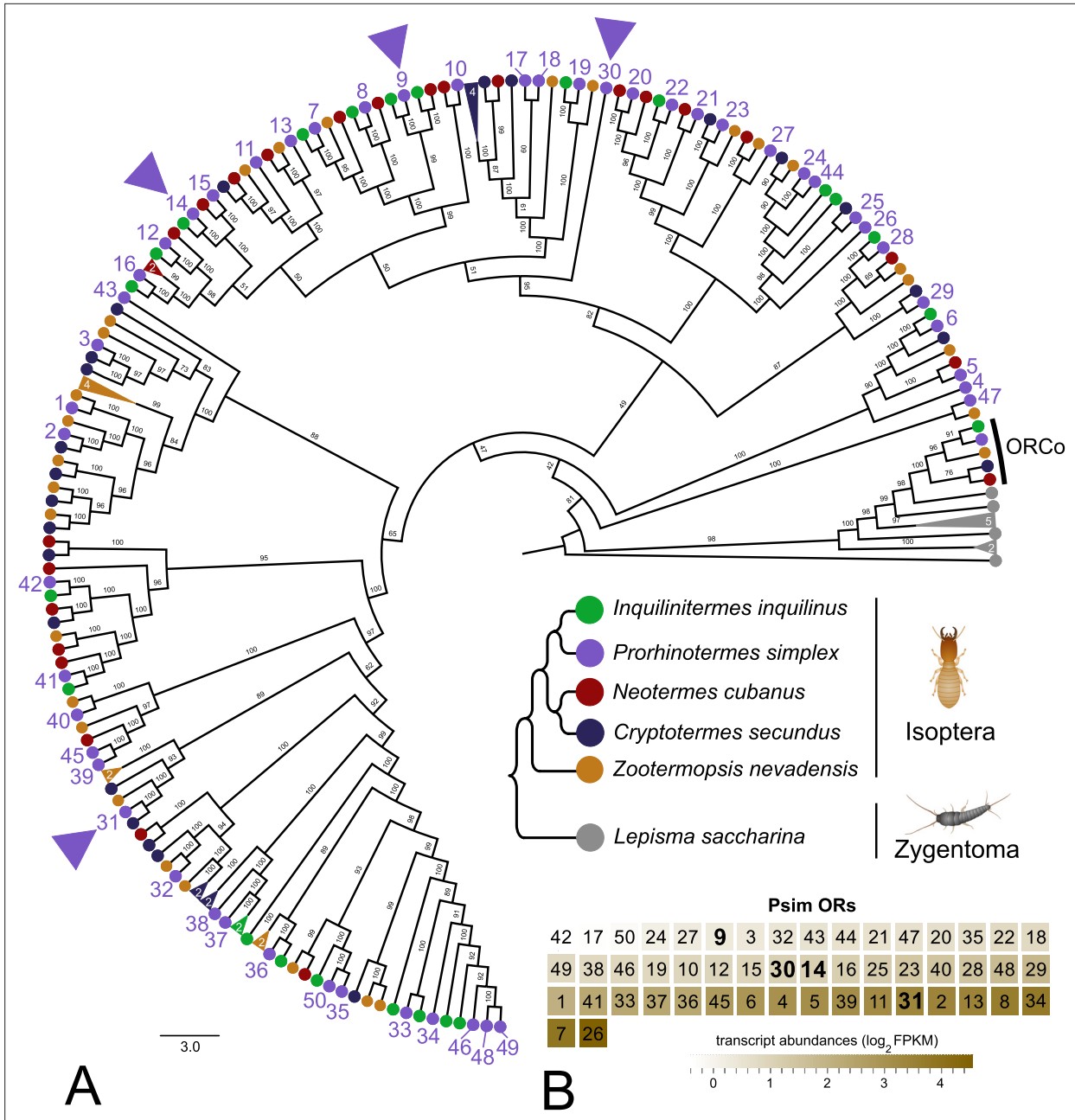

**Figure 1.** Phylogenetic reconstruction of termite odorant receptors (ORs) and their transcript abundances in *Prorhinotermes simplex* workers. (**A**) Phylogenetic tree is based on 182 protein sequences from five species of termites and the bristletail *Lepisma saccharina* as a basal insect outgroup, and also includes the sequences of ORCo. The topology and branching supports were inferred using the IQ-TREE maximum likelihood algorithm with the JTT+F+R8 model and supported by 10,000 iterations of ultrafast bootstrap approximation. Protein sequences of termite ORs can be found under the same labeling in *Johny et al., 2023*. *L. saccharina* sequences are listed in *Thoma et al., 2019*. Arrowheads highlight the four ORs from *P. simplex* selected for functional characterization. A fully annotated version of the tree is provided as *Figure 1—figure supplement 1*. (**B**) Heatmap shows the transcript abundances of 50 ORs identified in the RNAseq data from *P. simplex* worker antennae available in NCBI SRA archive under accession SRX17749141.

The online version of this article includes the following figure supplement(s) for figure 1:

**Figure supplement 1.** Fully annotated version of the phylogenetic tree of termite odorant receptors (ORs) shown in *Figure 1*.

Out of the 50 ORs identified in *P. simplex*, 26 sequences represented full open reading frames with at least 6 undisputed transmembrane domains predicted using TMHMM-2.0. From these, we selected four sequences (PsimOR9, 14, 30, and 31) situated in different parts (sub-branches) of the tree (*Figure 1*) and used them for transgenic *D. melanogaster* generation and SSR screening.

## Functional characterization of *P. simplex* ORs in *D. melanogaster* ab3 sensillum

We expressed the four selected *P. simplex* ORs in the recently improved version of the *D. melanogaster* Empty Neuron system *Chahda et al., 2019*; the crossing scheme for fly generation adapted

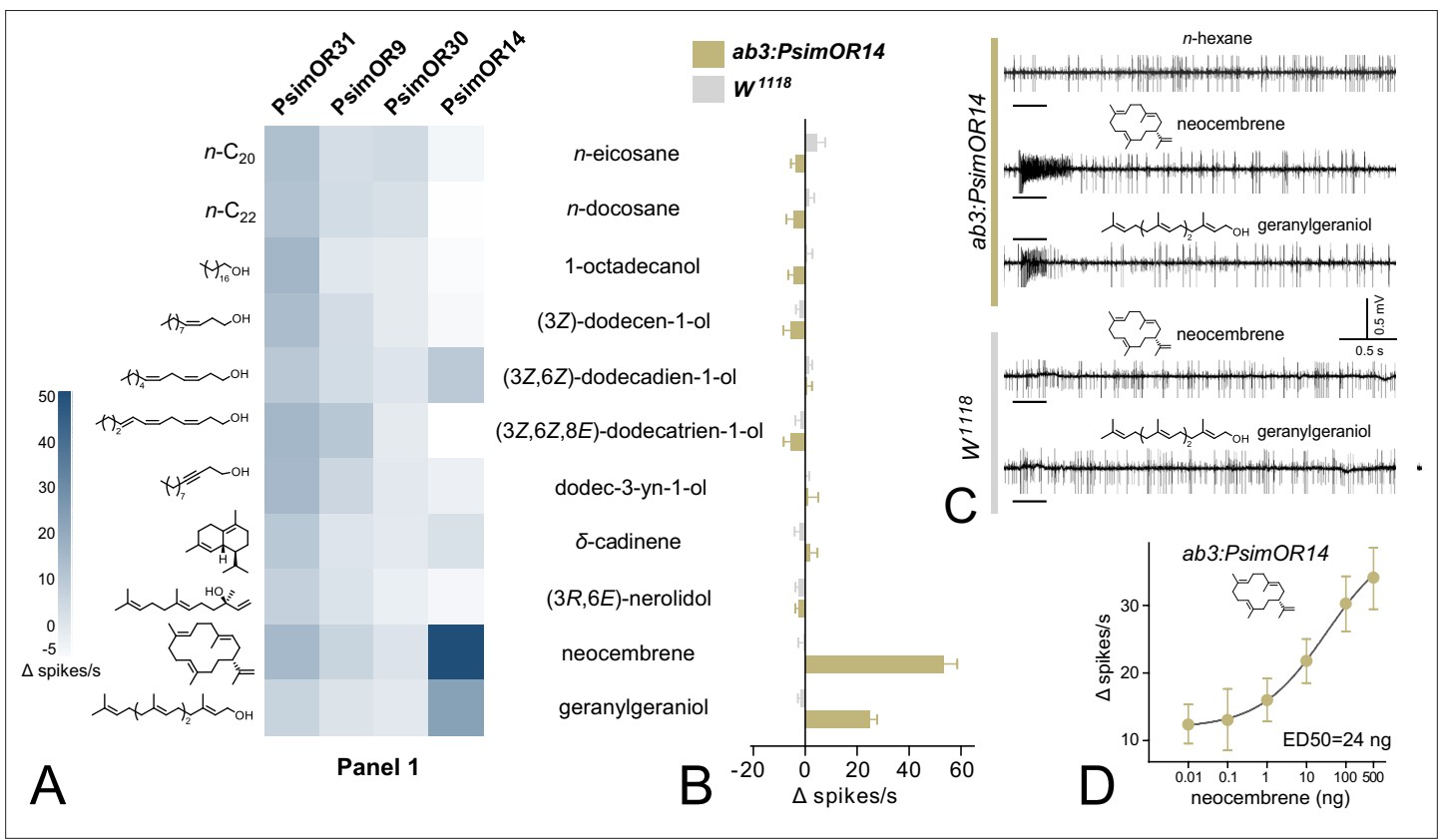

**Figure 2.** Single-sensillum recording (SSR) responses of transgenic *Drosophila melanogaster* ab3 sensillum expressing PsimOR9, 14, 30, and 31 to the initial screening of 11 volatiles with biological relevance for termites. (**A**) Heatmap showing the average responses of the four odorant receptors (ORs) as Δ spikes/s from 3 to 6 independent replicates. (**B**) Comparison of SSR responses of transgenic *D. melanogaster* ab3A neurons expressing PsimOR14 (*ab3A:PsimOR14*) and $W^{1118}$ *D. melanogaster*. The bars show the average Δ spikes/s values from five independent replicates ± SEM. (**C**) Characteristic SSR traces of *ab3A:PsimOR14* and $W^{1118}$ flies for 1 µg dose of neocembrene and geranylgeraniol. (**D**) Dose–response curve of *ab3A:PsimOR14* SSR responses to neocembrene. The graph shows average Δ spikes/s values ± SEM based on nine replicates (8 in case of 100 ng and 4 in case of 500 ng stimulations). The curve fit and ED50 value were calculated using log(agonist) versus response non-linear algorithm with least square fit method and the constraint of minimal response >0. The crossing scheme for transgenic fly generation is shown in *Figure 2—figure supplement 1*, the raw data for all graphs is provided in *Figure 2—source data 1–6*.

The online version of this article includes the following source data and figure supplement(s) for figure 2:

**Source data 1.** Single-sensillum recording (SSR) responses to Panel 1 for PsimOR9.

**Source data 2.** Single-sensillum recording (SSR) responses to Panel 1 for PsimOR14.

**Source data 3.** Single-sensillum recording (SSR) responses to Panel 1 for PsimOR30.

**Source data 4.** Single-sensillum recording (SSR) responses to Panel 1 for PsimOR31.

**Source data 5.** Single-sensillum recording (SSR) responses to Panel 1 for ab3:PsimOR14 versus $W^{1118}$.

**Source data 6.** Single-sensillum recording (SSR) dose–response data for neocembrene and PsimOR14 fly line.

**Figure supplement 1.** Crossing scheme of termite odorant receptors (ORs) heterologous expression using *Drosophila melanogaster* empty neurons in ab3 sensilla.

from *Gonzalez et al., 2016* is shown in *Figure 2—figure supplement 1*. Spontaneous SSR firing rates of the four transgenic lines showed an expected pattern with no abnormal bursts, indicating that the ORs were functional. The flies were first subjected to SSR screening with Panel 1, consisting of 11 semiochemicals relevant to termite chemical communication and structurally related compounds. As shown in *Figure 2A*, PsimOR9 and PsimOR30 did not provide any strong response to any of the tested compounds, and PsimOR31 broadly and weakly responded to several compounds. By contrast, PsimOR14 systematically and strongly responded to stimulations by the monocyclic diterpene hydro-carbon neocembrene, which is the main component of the TFP in the genus *Prorhinotermes* (*Sillam-Dussès et al., 2009*). Additionally, a moderate PsimOR14 response was also recorded for the linear diterpene alcohol geranylgeraniol, while all other compounds in the panel, including two other terpenoids, only elicited weak or no responses (*Figure 2A*).

We then compared the responses of *Drosophila* ab3 sensillum in PsimOR14 expressing flies with those of $W^{1118}$ flies. As evidenced in *Figure 2B*, the *W1118* ab3 sensillum did not show any significant neuronal response to Panel 1 compounds, while the transgenic PsimOR14 line generated an average Δ spike number of >50 spikes/s for neocembrene and a minor secondary response of ~25 Δ spikes/s for geranylgeraniol. Characteristic responses for both lines to the two compounds are depicted in *Figure 2C*. In the next step, we tested the dose–response behavior of PsimOR14 flies to neocembrene

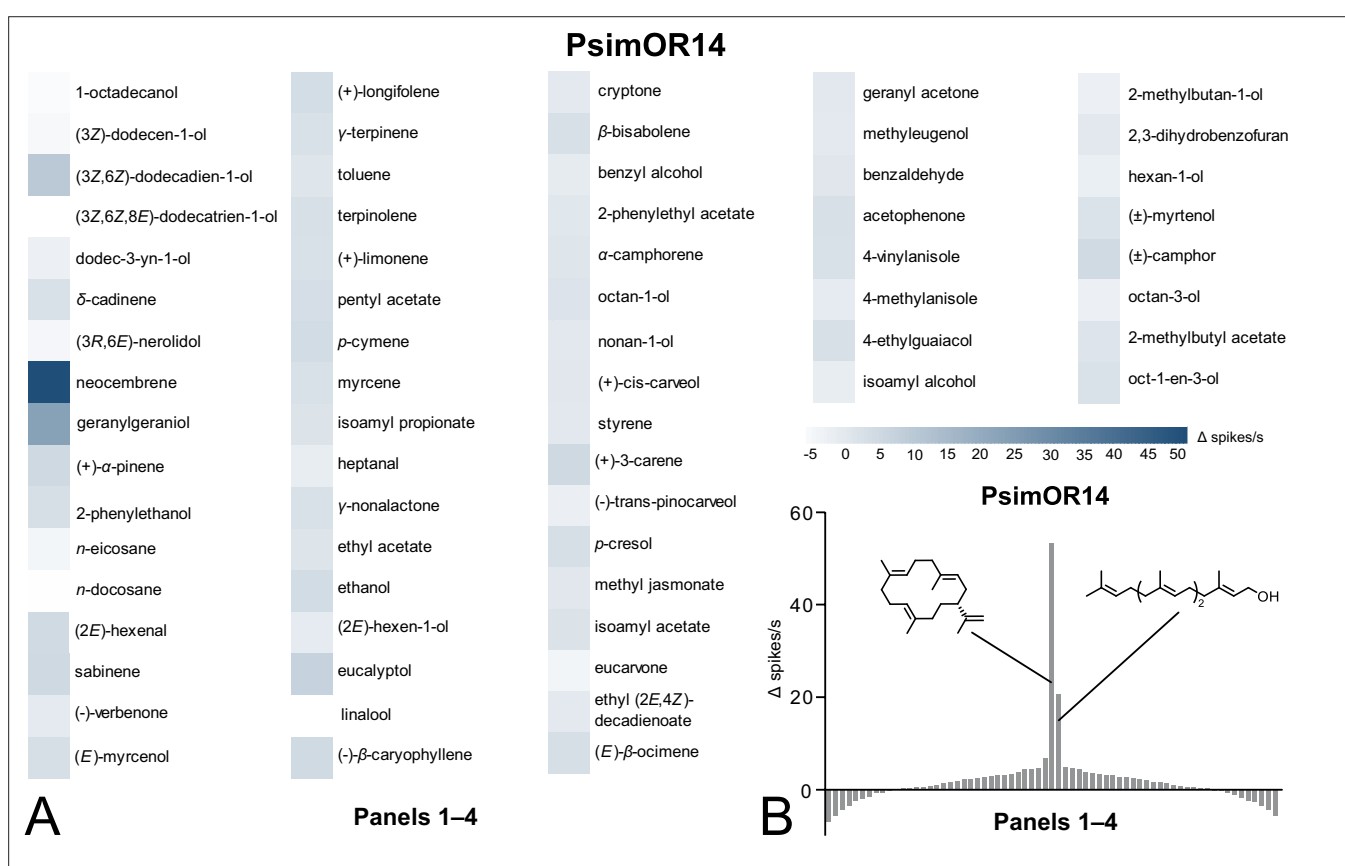

**Figure 3.** Single-sensillum recording (SSR) responses of transgenic *D. melanogaster* ab3 sensillum expressing PsimOR14 to the complete set of 67 compounds (Panels 1–4). (**A**) Heatmap showing the average responses as Δ spikes/s from 3 to 6 independent replicates. (**B**) Tuning curve of PsimOR14 for the 67 compounds contained in Panels 1–4. The raw data for both graphs is provided in *Figure 3—source data 1–3*. Origin and purity of the tested chemicals are provided in *Figure 3—source data 4*.

The online version of this article includes the following source data for figure 3:

**Source data 1.** Single-sensillum recording (SSR) responses to Panel 2 for PsimOR14.

**Source data 2.** Single-sensillum recording (SSR) responses to Panel 3 for PsimOR14.

**Source data 3.** Single-sensillum recording (SSR) responses to Panel 4 for PsimOR14.

**Source data 4.** Origin and purity of the tested chemicals.

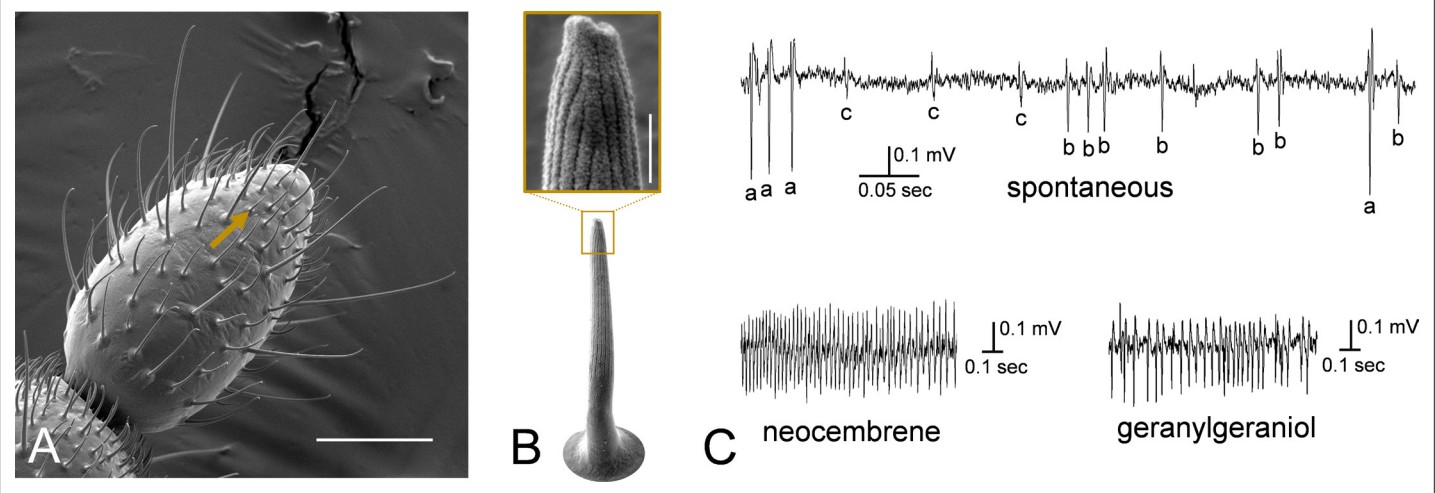

**Figure 4.** Neocembrene-responding sensillum in *P. simplex* workers. (**A**) Scanning electron microscopy (SEM) photograph of the last flagellomere of *P. simplex* worker. Arrow shows a small multiporous grooved sensillum responding to neocembrene and geranylgeraniol. Scale bar represents 50 µm. (**B**) High-resolution SEM (HR-SEM) view on the neocembrene-responding sensillum. Scale bar in the inset represents 500 nm. (**C**) Detailed view on single-sensillum recording (SSR) traces recorded from the neocembrene-responding sensillum during spontaneous firing, and upon stimulation with neocembrene and geranylgeraniol.

and recorded an exponentially increasing neuronal response over the range of 0.01–10 ng to ED50 = ~24 ng and a lack of saturation at the dose of 500 ng (*Figure 2D*).

## PsimOR14 is narrowly tuned to neocembrene, the main TFP component

To further address the specificity of PsimOR14 tuning, we tested three additional panels containing 56 frequently occurring insect semiochemicals from various chemical classes. As shown in *Figure 3A*, none of these compounds, including multiple terpenoids (mono-, sesqui-, di-), generated a strong response, suggesting a narrow tuning of PsimOR14 to neocembrene. The narrow tuning of PsimOR14 is evident also from the tuning curve depicted in *Figure 3B*, with the receptor lifetime sparseness value 0.88. These results confirm that PsimOR14 is a pheromone receptor adaptively tuned to detect the main TFP component, neocembrene.

## Identification of *P. simplex* olfactory sensillum responding to neocembrene

Our next goal was to identify the antennal olfactory sensillum responsible for neocembrene detection by *P. simplex* workers using a combination of SSR measurements with scanning electron microscopy (SEM) and high-resolution SEM (HR-SEM) imaging. Since no previous study reported SSR responses of termite olfactory sensilla to pheromones or environmental cues, we decided to search for the neocembrene-detecting sensillum on the last flagellomere, known to harbor by far the most sensilla in termite workers (*Castillo et al., 2021*).

In SSR experiments with the termite-relevant compounds from Panel 1, we obtained strong response to both neocembrene and geranylgeraniol from a short multiporous grooved sensillum situated in the apical part of the last antennal segment (*Figure 4A, B*). The response of this sensillum to neocembrene was confirmed on workers originating from two different colonies. Detailed view on spontaneous firing pattern of the neocembrene-responding sensillum revealed three different spike amplitudes (*Figure 4C*), suggesting the potential presence of as many as three olfactory sensory neurons (a–c). Comparison with spike amplitudes upon neocembrene and geranylgeraniol stimulations then indicated that the responses are generated by the neuron labeled as b.

The SSR response spectrum of neocembrene sensillum to Panel 1 was markedly similar to that of ab3A neuron of PsimOR14-expressing *Drosophila* (*Figure 5A*). None of the compounds elicited higher average responses than 10 Δ spikes/s, except for neocembrene and geranylgeraniol; their average Δ spikes/s were even slightly higher than those of heterologously expressed PsimOR14,

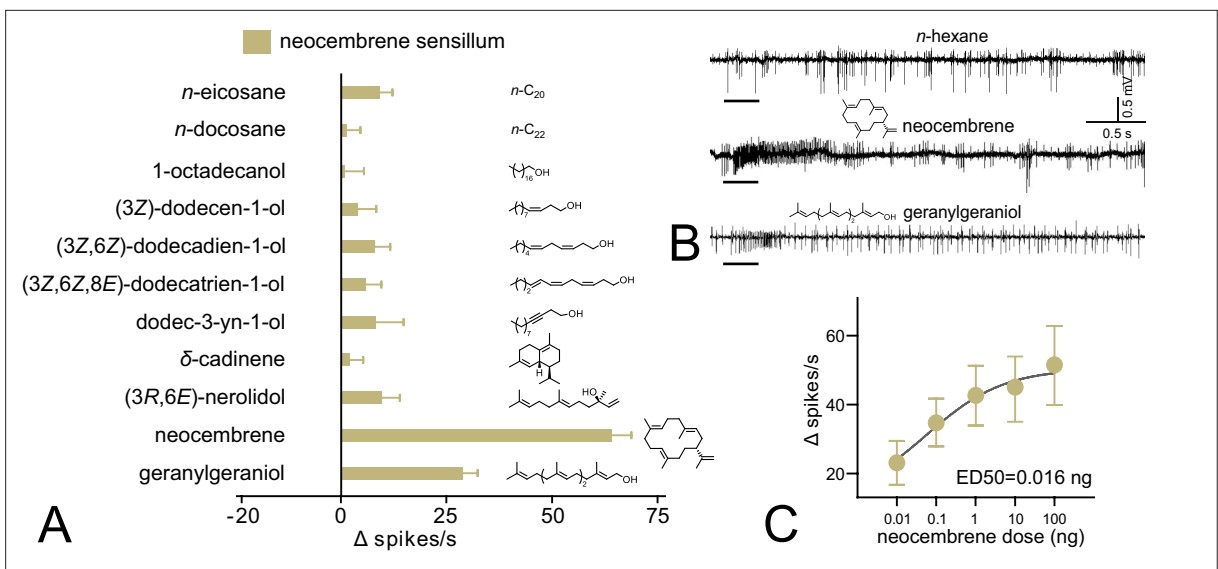

**Figure 5.** Single-sensillum recording (SSR) responses of the neocembrene-responding sensillum on the last flagellomere of *P. simplex* worker. (**A**) SSR responses to Panel 1. The bars show the average Δ spikes/s values from 8 to 17 replicates ± SEM. The raw data is provided in ***Figure 5—source data 1***. (**B**) Characteristic SSR traces of the neocembrene-detecting sensillum for neocembrene and geranylgeraniol. (**C**) Dose–response curve of the SSR responses to neocembrene by the neocembrene-responding sensillum. The graph shows average Δ spikes/s values ± SEM based on 9–11 replicates. The curve fit and ED50 value were calculated using log(agonist) versus response non-linear algorithm with least square fit method and the constraint of minimal response >0. The raw data is provided in ***Figure 5—source data 2***.

The online version of this article includes the following source data for figure 5:

**Source data 1.** Single-sensillum recording (SSR) responses to Panel 1 by neocembrene sensillum in *P. simplex* workers.

**Source data 2.** Single-sensillum recording (SSR) dose–response data for neocembrene and *P. simplex* neocembrene sensillum.

reaching ~65 and ~29, respectively (***Figure 5A, B***). Likewise, the dose–response experiment with neocembrene indicated a higher sensitivity threshold and lower ED50 = 0.016 ng (***Figure 5C***).

## PsimOR14 gene and protein structure, protein modeling, ligand docking, MM/PBSA, MD simulations

Mapping the *PsimOR14* transcript sequence on *P. simplex* draft genome revealed that the gene consists of six exons and is situated on the same locus and in close vicinity of *PsimOR15*, with which it shares the exon-intron boundaries, suggesting a recent diversification of the two genes via duplication, as also supported by their high sequence similarity (***Figure 1A*** and ***Figure 6A***). Transcript (***Figure 6B***) and protein (***Figure 6C***) structures of PsimOR14 showed the presence of seven transmembrane domains (S1–S7) with the largest extracellular loop between S3 and S4 and the longest intracellular loop between S4 and S5.

  ***Figure 6D*** shows the initial PsimOR14 model obtained using AlphaFold 2. Three terpenoid ligands, that is, the best agonists neocembrene and geranylgeraniol, and a weak agonist (+)-limonene, were selected for docking into the identified binding site. Dockings are visualized in ***Figure 6E***, the final scores of the best-ranked poses are shown in ***Table 1***. The predicted docking scores indicated neocembrene as the best ligand, followed by geranylgeraniol and (+)-limonene, in line with the ranking of their biological effect in SSR assays (***Table 1***). (+)-Limonene ranked as the worst agonist also according to the binding free energy calculated in MM/PBSA analysis, while the best energy score was obtained for geranylgeraniol followed by neocembrene (***Table 1***). Both docking and MM/PBSA analysis suggested that primarily Van der Waals interactions facilitate the binding; only in the case of geranylgeraniol has a non-negligible contribution of electrostatic interactions been recorded. Per-residue decomposition results (***Figure 6C***) showed that all three ligands bind two hydrophobic patches made out of residues from S3 and S4 (Cys154, Ala157, Val158; Thr221, Leu224, Ala225, Tyr228). Neocembrene and (+)-limonene only bind these patches, while geranylgeraniol also interacts with additional residues (***Figure 6—source data 2***).

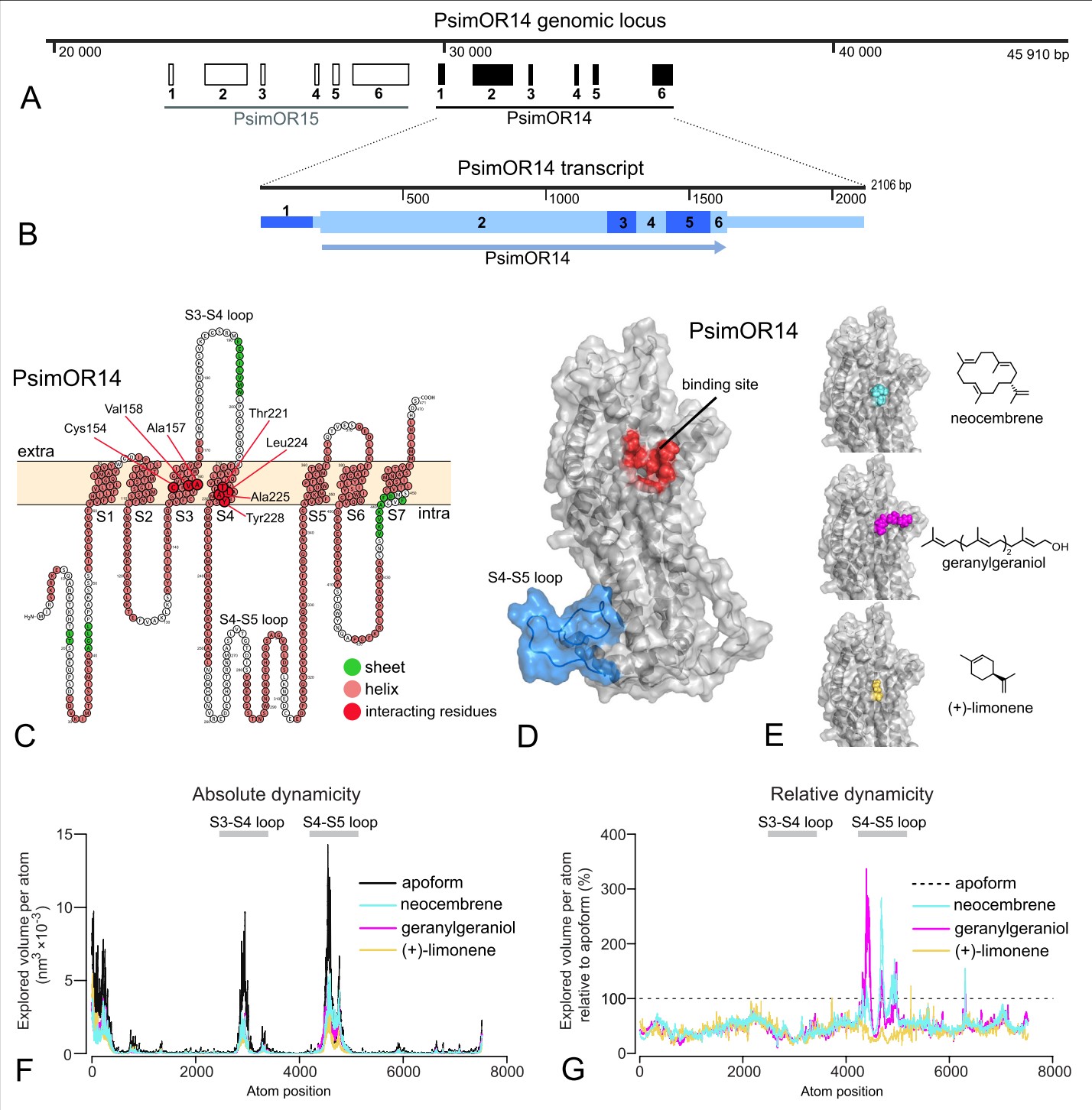

**Figure 6.** PsimOR14 gene, transcript, and protein structures, docking, and molecular dynamics (MD) simulations. (**A**) Genomic locus containing *PsimOR14* and *PsimOR15*. *PsimOR14* gene consists of one non-coding and five protein-coding exons. (**B**) *PsimOR14* transcript with six exons, showing the protein-coding (higher boxes) and untranslated regions (lower boxes), and open reading frame (ORF; arrow). (**C**) Transmembrane architecture of PsimOR14. In red are shown seven residues interacting with neocembrene. Light blue ellipse shows the intracellular loop the most impacted by ligand binding. (**D**) Modeled apoform of PsimOR14. Red region denotes the binding site identified via docking, light blue region represents the intracellular S4–S5 loop. (**E**) Holoforms of PsimOR14 with three docked ligands. (**F**) Absolute PsimOR4 dynamicity expressed as average volumes explored by atoms per simulation step in PsimOR14 apoform and upon binding the three studied ligands. (**G**) Relative PsimOR14 dynamicity expressed as average explored atom volumes upon ligand binding relative to the volumes in PsimOR14 apoform. Nucleotide and protein sequences of PsimOR14 are

*Figure 6 continued on next page*

*Figure 6 continued*

provided under NCBI entry OR921181 and as *Figure 6—source data 1*. Interacting residues and their binding energies to the ligands are listed in *Figure 6—source data 2*. The raw data for explored atomic volumes are provided in *Figure 6—source data 3*.

The online version of this article includes the following source data for figure 6:

**Source data 1.** Nucleotide and amino acid sequences of PsimOR14.

**Source data 2.** Residues interacting with the docked ligands and their binding energies.

**Source data 3.** Explored atomic volumes of PsimOR14 apoform and upon binding of ligands, inferred from molecular dynamics simulations.

MD simulations of per atom explored volumes in PsimOR14 apoform and upon ligand binding delimited three regions with high dynamicity, that is the N-terminal region, the extracellular S3 and S4 loops, and especially the intracellular S4 and S5 loop (*Figure 6F*). Binding of each of the three ligands reduced the overall protein dynamicity; this protein stabilization did not differ dramatically among the three ligands. By contrast, binding of geranylgeraniol and neocembrene led to a conspicuous dynamicity increase in a portion of the S4 and S5 loop, compared to both the apoform and (+)-limonene binding (*Figure 6G*).

## Caste-biased PsimOR14 expression and antennal sensitivity to neocembrene

We next decided to compare the expression pattern of PsimOR14 between *P. simplex* workers and soldiers, along with the sensitivity of the two castes to its preferred ligand, neocembrene. Both DESeq2 and EdgeR differential expression analyses of RNAseq read counts from heads (including antennae) of workers and soldiers revealed that PsimOR14 is significantly more expressed in workers, being among the three most upregulated ORs in workers (*Figure 7A*). Subsequent electroantennographic (EAG) measurements were in line with this observation and indicated significantly stronger responses to neocembrene in workers (p = 0.012) (*Figure 7B*).

## Discussion

Identification of PsimOR14 as the pheromone receptor narrowly tuned to neocembrene in *P. simplex* represents the first OR deorphanization in termites and confirms that the trail-following communication is mediated by ORs. This assumption, validated here for the monocyclic diterpene neocembrene, is indirectly supported by previous experiments in two termite species having a $C_{12}$ fatty alcohol as TFP; ORCo silencing in these species impaired the ability to follow the foraging trail (*Gao et al., 2020*). Because trail following in termites has the shared evolutionary origin with courtship communication and both neocembrene and $C_{12}$ alcohols also occur as sex-pairing pheromone components (*Bagnères and Hanus, 2015*; *Bordereau and Pasteels, 2011*; *Sillam-Dussès, 2010*), it is likely that the two communication modalities share identical or closely related ORs. Future OR deorphanizations should test this hypothesis and provide an insight on how the functional diversification of trail following and sex attraction is imprinted into the OR evolution.

Neurophysiological characteristics of *D. melanogaster* ab3A neuron expressing PsimOR14 showed expected patterns of spontaneous firing rates of units of Hz (6.87 ± 4.73, mean ± SD) and maximum firing rates of 90 spikes/s at the highest used neocembrene doses (1 µg), though not reaching the reported maxima for ab3A responses with genuine and exogenous receptors, which

**Table 1.** Docking scores and energy values inferred from the docking experiment and from MM/PBSA simulations for binding interactions of neocembrene, geranylgeraniol, and (+)-limonene with PsimOR14.

| Ligand | Docking experiment | | | MM/PBSA E (kcal/mol) ± SD | | | |
|---|---|---|---|---|---|---|---|
| | Docking score | VDWAALS | Electrostatic | ΔTOTAL | ΔVDWAALS | ΔEEL | ΔGSOLV |
| | | (kcal/mol) | | | | | |
| Neocembrene | −8.658 | −19.777 | −0.223 | −28.72 ± 1.46 | −26.92 ± 1.24 | −0.29 ± 0.56 | −1.51 ± 0.10 |
| Geranylgeraniol | −8.331 | −18.786 | −11.137 | −36.98 ± 1.22 | −35.47 ± 0.96 | −0.77 ± 0.56 | −0.73 ± 0.27 |
| (+)-Limonene | −7.638 | −16.134 | −0.561 | −20.02 ± 2.63 | −20.57 ± 2.30 | −0.35 ± 0.47 | 0.89 ± 2.21 |

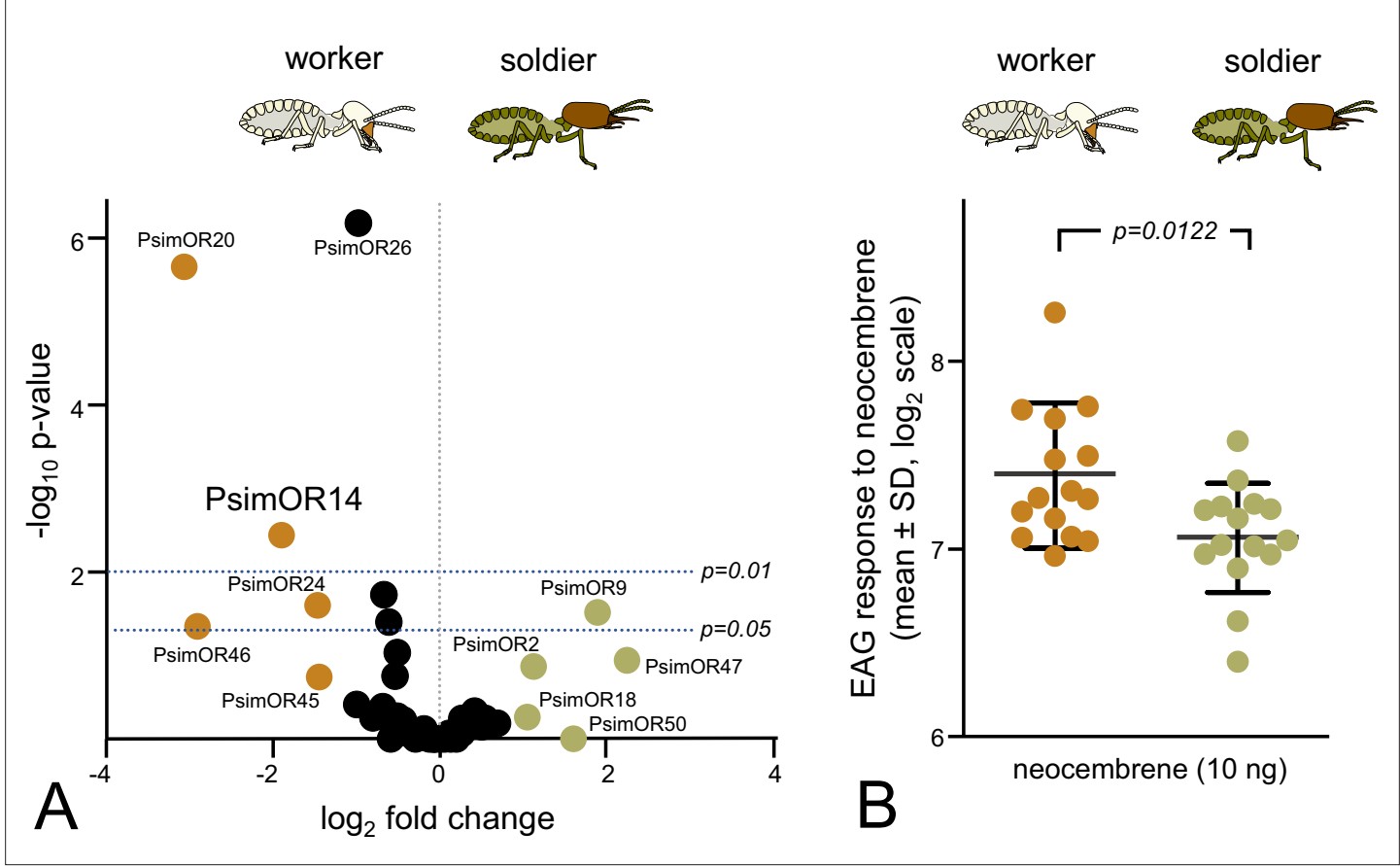

**Figure 7.** Caste comparison of *PsimOR14* expression and EAG responses between *P. simplex* workers and soldiers. (**A**) Volcano plot representing edgeR differential gene expression analysis of all 50 *P. simplex* odorant receptors (ORs) in RNAseq data from soldier and workers heads (including antennae) sequenced in three independent biological replicates per caste. Colored dots mark ORs reaching absolute value of $\log_2$ fold change $\geq 1$, horizontal lines represent p-value thresholds of 0.05 and 0.01. Based on SRA archives accessible under SRX18952230–32 and SRX18952237–39. Numeric values of the edgeR and DESeq2 differential expression analysis are provided in *Figure 7—source data 1*. (**B**) EAG responses of whole antenna preparations of workers and soldiers to neocembrene at a dose of 10 ng (mean ± SD shown on $\log_2$ scale). Inter-caste differences were compared using a *t*-test on $\log_2$-transformed data. Raw data is shown in *Figure 7—source data 2*.

The online version of this article includes the following source data for figure 7:

**Source data 1.** Results of the edgeR and DESeq2 differential expression analyses.

**Source data 2.** EAG measurements.

may be well over 100 spikes/s (e.g., *Chahda et al., 2019*). This confirms that co-expression of termite ORs with the *Drosophila* ORCo using the Empty Neuron system is a suitable technique for termite OR characterizations. PsimOR14 has a narrow tuning to neocembrene with receptor lifetime sparseness equal to 0.88. ORs detecting pheromone components (pheromone receptors) and other volatiles with high biological importance, such as key food or host attractants, are often expressed in specialized sensilla and are highly selective, in contrast to the broad tuning of ORs sensing the general environmental stimuli (e.g., *Carey et al., 2010*; *Fleischer and Krieger, 2018*). Such a high selectivity is the best known for sex pheromone receptors, for example in moths (reviewed in *Zhang and Löfstedt, 2015*) or *Drosophila* (reviewed in *Haverkamp et al., 2018*), mostly tuned to a single compound, even though in some cases the respective ORs can be adaptively shaped to selectively detect more pheromone compounds (*Díaz-Morales et al., 2024*; *Mariette et al., 2024*). Thus, the narrow tuning of PsimOR14 is in line with expectations from a pheromone receptor, especially when considering the ancient origin of trail-following from sex-pairing behavior and the great importance of TFP for concerted foraging in these blind insects. Nevertheless, due to the low coverage of insect diversity by OR deorphanization studies and their strong bias toward derived

Holometabola, it remains difficult to make any general considerations about the OR selectivity relative to the receptor function across Insecta. This has been recently demonstrated by a comprehensive functional characterization of ORs in migratory locusts, unveiling an unexpected design of the olfactory information decoding, consisting of a large set of narrowly tuned ORs for environmental cues (*Chang et al., 2023*).

With average Δ spike/s over 20, the linear diterpene alcohol geranylgeraniol was the only other agonist with non-negligible response. None of the remaining compounds elicited a Δ spike of more than 7, despite the presence of multiple other compounds derived from terpenoid scaffolds in the tested panels. It is difficult to attribute any adaptive significance to this observation since there is no record of geranylgeraniol in communication context in termites. Therefore, the observed significant biological activity, high docking score, and high ranking in free energy comparison may be due to non-adaptive binding affinity of PsimOR14 to the non-native ligand (see also below). Interestingly, another related terpene alcohol, the monoterpene linalool, had the biggest negative Δ spike score (–6.8) and the absolute spikes/s ranged from 0 to 5 (1.5 ± 1.9, mean ± SD), suggesting a possible inverse agonist function of linalool.

The neocembrene-sensing sensilla identified using SSR in *P. simplex* workers provided a response pattern to Panel 1 very similar to that of *PsimOR14*-expressing ab3 *Drosophila* sensillum, for example, strong response to neocembrene followed by geranylgeraniol, supporting independently the results obtained from transgenic *Drosophila*. No other important responses to termite pheromone components from the Panel 1, which also included the minor *P. simplex* TFP component (3*Z*,6*Z*,8*E*)-dodecatrien-1-ol, were recorded, in spite of the likely presence of two additional olfactory neurons housed in the neocembrene-detecting sensillum. The function of these additional neurons remains elusive, and examples from other insects offer multiple scenarios. A neuron expressing a highly selective pheromone receptor may co-exist with functionally independent neurons detecting environmental cues (e.g., *Tateishi et al., 2020*), or the co-habitation of several neurons may generate more complex interactions such as lateral inhibition of the neighboring neurons (*Su et al., 2012*; *Zhang et al., 2019b*). However, it must be noted that our SSR analyses focused on a specific sensillum defined by its topology on the last flagellomere, and we did not perform a comprehensive mapping of olfactory sensilla on the entire worker antenna. Therefore, our image on the distribution of neocembrene-detecting sensilla and the pattern of neurons housed in these sensilla is incomplete.

The basic PsimOR14 protein architecture with seven transmembrane domains is typical for insect ORs. Likewise, the docking experiments identified a binding site defined by a binding pocket deep in the transmembrane region, homologous to that in previously studied insect ORs, and confirmed the nature of the ligand binding to mainly rely on hydrophobic interactions (e.g., *Del Mármol et al., 2021*; *Pettersson and Cattaneo, 2023*; *Yuvaraj et al., 2021*). In the case of PsimOR14, mainly the residues from S3 and S4 interacted with the studied ligands; these transmembrane domains were shown to participate in ligand binding also in other studied insect ORs (e.g., *Del Mármol et al., 2021*; *Yuvaraj et al., 2021*; *Wang et al., 2024*; *Zhao et al., 2024*).

The calculated binding scores for the three tested ligands correlated with observed biological effects of the ligands. The MM/PBSA data also in part corroborated the SSR measurements by estimating (+)-limonene as the worst agonist; by contrast, geranylgeraniol and not neocembrene ranked as the best ligand. This may potentially be explained by the additional residues interacting with geranylgeraniol compared to neocembrene and (+)-limonene, and the relatively higher contribution of electrostatic interactions in geranylgeraniol binding. Moreover, in systems where ligand binding induces allosteric effects, as is the case of insect ORs, the simple binding affinity does not provide a complete picture of receptor signal transduction function.

In contrast to the general protein stabilization upon binding of the three tested ligands, some parts of the intracellular S4 and S5 loops showed a significant increase in the dynamics upon binding of the biologically active ligands geranylgeraniol and neocembrene, compared to apoform and the weak agonist (+)-limonene binding. What is the mechanistic role of this allosteric transmission of the binding effect to the intracellular loop remains elusive. However, the impact of the S4-S5 loop on OR function has previously been reported for the MhOR5 from *Machilis hrabei Del Mármol et al., 2021*; replacement of the loop with a short linker increased the receptor response to native ligand. Interestingly, in some insect ORs, this loop has been reduced to a short sequence of a few residues (e.g., *Wang et al., 2024*).

*PsimOR14* does not belong to the most expressed *P. simplex* ORs (*Figure 1B*). However, interestingly, it is among those having the most caste-biased expression with significantly higher transcript abundance in antennae of workers compared to soldiers. Accordingly, the electrophysiological responses of workers to neocembrene were significantly stronger in workers. Termite soldiers are known to lay pheromone trails, to detect TFPs and to participate in foraging and field exploration in a number of termite species, though the behavioral patterns of soldiers and workers during these activities differ (*Kaib, 1990*; *Traniello, 1981*; *Traniello and Busher, 1985*). This has also been demonstrated in a close relative to our model, the congeneric species *Prorhinotermes inopinatus* (*Rupf and Roisin, 2008*). Yet, differences in sensitivity of termite workers and soldiers have not previously been addressed at the electrophysiological level. Caste-biased *PsimOR14* expression and neocembrene sensitivity may represent olfactory information filtering adaptive to the different tasks of the two castes, as documented, for example, in ants (*Caminer et al., 2023*).

Genus *Prorhinotermes* is the most basally situated termite taxon known to have a terpenoid component as a part of its pheromone repertoire (*Sillam-Dussès et al., 2009*). The acquisition of terpenoid pheromone components is undoubtedly due to the evolution of terpene biosynthesis in basal Neoisoptera, which has led to the fascinating diversity of defensive terpenoids produced by soldiers of Neoisoptera in their frontal gland (*Gössinger, 2019*). In line with general observations on pheromone evolution (*Steiger et al., 2011*), some of these defensive terpenoids from the frontal gland gained via exaptation a novel function of alarm pheromones (*Dolejšová et al., 2014*; *Roisin et al., 1990*; *Šobotník et al., 2008*; *Šobotník et al., 2010*). Likewise, the cyclic diterpenes probably became part of TFPs and sex-pairing pheromones by co-opting the terpenoid biosynthesis in exocrine glands other than the defensive frontal gland of soldiers. Beside the occurrence of neocembrene in the sub-basal *Prorhinotermes*, it only occurs as a pheromone component in much later diverging representatives of the subfamily Nasutitermitinae (*Bordereau and Pasteels, 2011*). Once reliable genome or antennal transcriptome of Nasutitermitinae becomes available, it would be of interest to search in these species for OR orthologs of PsimOR14 and test whether they have retained the function of neocembrene detection.

Insect ORs are frequently organized in tandem arrays (e.g., *Bohbot et al., 2007*; *McKenzie and Kronauer, 2018*; *Robertson and Wanner, 2006*). Likewise, the genomic locus of *PsimOR14* contains a likely tandem copy paralog of *PsimOR15* gene. The two genes share the gene architecture, and their transcripts are equally represented in worker antennal transcriptome, though *PsimOR15* does not display the caste-biased expression. Thanks to the close similarity of both genes, *PsimOR15* is a candidate for potential pheromone receptor function.

Future research in termites should also aim at finding the ORs or other chemosensory proteins involved in the detection of CHCs. In spite of the similarities in CHC roles in termites and ants, ranging from nestmate recognition to fertility signaling, termite OR sequences do not show any conspicuous expansions analogous to 9-exon subfamily in ants. Therefore, finding the chemosensory principle of CHC detection in termites would bring another piece of knowledge about the fascinating convergent evolution in these two unrelated major groups of eusocial insects. Yet another appealing target for OR deorphanization is the search for pheromone receptors of queen pheromones, the central signals ensuring the reproductive dominance of queens in the colonies of social insects. In the honeybee and ants, narrowly tuned ORs responding to main queen pheromone components were already described (*Wanner et al., 2007*; *Pask et al., 2017*). In termites, queen pheromones were so far only identified in two species (*Dolejšová et al., 2022*; *Matsuura et al., 2010*). They are volatile and were shown to act as airborne signals via olfaction (*Dolejšová et al., 2022*); it is thus reasonable to expect a selective OR responsible for their detection.

# Materials and methods

**Key resources table**

| Reagent type (species) or resource | Designation | Source or reference | Identifiers | Additional information |
|---|---|---|---|---|
| Sequence-based reagent | PsimOR14_F | This paper | PCR primer – cloning | ATGATTCGATCAAAGAGAAAGG |
| Sequence-based reagent | PsimOR14_R | This paper | PCR primer – cloning | TTAGGAGTCGTGTAGATGAAT |
| Sequence-based reagent | PsimO31_F | This paper | PCR primer – cloning | ATGGAATACATAAAAAATGAAACATATTCTCA |
| Sequence-based reagent | PsimO31_R | This paper | PCR primer – cloning | TCAACCTACGACATGTGAGTTATT |
| Sequence-based reagent | PsimOR9_F | This paper | PCR primer – cloning | ATGGACAGCCTTTACGACCAATCTT |
| Sequence-based reagent | PsimOR9_R | This paper | PCR primer – cloning | TCATTCAGTGACTGAGGGATCCTT |
| Sequence-based reagent | PsimO30_F | This paper | PCR primer – cloning | ATGGAGCACAGGAAATACAAAGTGACAA |
| Sequence-based reagent | PsimO30_R | This paper | PCR primer – cloning | TTACGTTCCCTGATTTGTGTCGGTAT |
| Sequence-based reagent | PsimOrco_F | This paper | PCR primer – cDNA check | ATGTACAAGTTCAGGTTACACG |
| Sequence-based reagent | PsimOrco_R | This paper | PCR primer – cDNA check | CTAGTTGAGCTGTACCAACAC |
| Sequence-based reagent | GW1 | Thermo Fisher Scientific | PCR and Sanger sequencing primer | GTTGCAACAAATTGATGAGCAATGC |
| Sequence-based reagent | GW2 | Thermo Fisher Scientific | PCR and Sanger sequencing primer | GTTGCAACAAATTGATGAGCAATTA |
| Sequence-based reagent | UAS1 | *Gonzalez et al., 2016* | Sanger sequencing primer | TAGCGAGCGCCGGAGTATAAATAG |
| Sequence-based reagent | UAS2 | *Gonzalez et al., 2016* | Sanger sequencing primer | ACTGATTTCGACGGTTACCC |
| Sequence-based reagent | DmOr22a_F | This paper | PCR primer – genotyping | TCTCCAGCATCGCCGAGTGT |
| Sequence-based reagent | DmOr22a_R | This paper | PCR primer – genotyping | CGGCAGAGGTCCAGTCCGAT |
| Sequence-based reagent | PsimOR14_SW_F | This paper | PCR primer – genotyping | GAGAGCCAAGCAAACGAAAC |
| Sequence-based reagent | PsimOR14_SW_R | This paper | PCR primer – genotyping | TTTAGAAGGGAGCCACATCAC |
| Sequence-based reagent | PsimO31_SW_F | This paper | PCR primer – genotyping | GCTGGGTTAATCCCGATCAT |
| Sequence-based reagent | PsimO31_SW_R | This paper | PCR primer – genotyping | GCATGGCACCAAATAGTTCTTC |
| Sequence-based reagent | PsimOR9_SW_F | This paper | PCR primer – genotyping | TGGGCGAAACTGAGGATATG |
| Sequence-based reagent | PsimOR9_SW_R | This paper | PCR primer – genotyping | CGAGCCGACATAGAAGAAGAG |
| Sequence-based reagent | PsimO30_SW_F | This paper | PCR primer – genotyping | TGCCATCACCAGCAGATAAA |
| Sequence-based reagent | PsimO30_SW_R | This paper | PCR primer – genotyping | CACCGACTGACTCAGCATATT |

*Continued on next page*

*Continued*

| Reagent type (species) or resource | Designation | Source or reference | Identifiers | Additional information |
|---|---|---|---|---|
| Commercial assay or kit | PureLink RNA Mini | Invitrogen | Cat. #: 12183018A | |
| Commercial assay or kit | SuperScript IV Reverse Transcriptase | Invitrogen | Cat. #: 18090050 | |
| Commercial assay or kit | DreamTaq Green PCR Master Mix | Invitrogen | Cat. #: K1081 | |
| Commercial assay or kit | QIAquick Gel Extraction Kit | QIAGEN | Cat. #: 28706 | |
| Commercial assay or kit | pCR8/GW/TOPO TA Cloning Kit | Invitrogen | Cat. #: K250020 | |
| Strain, strain background (*Escherichia coli*) | OneShot TOP10 | Invitrogen | Cat. #: C404010 | Competent cells, Certificates of Analysis available at https://www.thermofisher.com/order/catalog/product/C404010 |
| Commercial assay or kit | QIAprep Spin Miniprep Columns | QIAGEN | Cat. #: 27115 | |
| Commercial assay or kit | Gateway LR Clonase Enzyme mix | Invitrogen | Cat. #: 11791019 | |
| Recombinant DNA reagent | pUASg.attb (plasmid) | *Drosophila* Genomics Resource Center, Bloomington, USA | DGRC Stock 1422; https://dgrc.bio.indiana.edu//stock/1422; RRID:DGRC_1422 | |
| Recombinant DNA reagent | pUASg.attB-PsimOR (plasmid) | This paper | | |
| Genetic reagent (*D. melanogaster*) | w; Or22ab$^{GAL4}$ | Thomas O. Auer (from Richard Benton Lab, University of Lausanne, Switzerland) | FLYB:FBal0018186 | *Chahda et al., 2019* |
| Genetic reagent (*D. melanogaster*) | W$^{1118}$ | Michal Žurovec (from Laboratory of Molecular Genetics, Institute of Entomology, Czechia) | | |
| Genetic reagent (*D. melanogaster*) | w-; Bl/Cyo; TM2/TM6B | MPI-Jena, Germany | | |
| Genetic reagent (*D. melanogaster*) | w-; +/+; UAS-OR(w+)/ UAS-OR(w+) | This paper | | On-demand commercial transgenesis by BestGene Inc, USA |
| Sequence-based reagent (*Cryptotermes secundus*) | CsecOR and ORco sequences | *Johny et al., 2023* | | |
| Sequence-based reagent (*Zootermopsis nevadensis*) | ZnevOR and ORco sequences | *Johny et al., 2023* | | |
| Sequence-based reagent (*Lepisma saccharina*) | LsacOR sequences | *Thoma et al., 2019* | | |

*Continued on next page*

*Continued*

| Reagent type (species) or resource | Designation | Source or reference | Identifiers | Additional information |
|---|---|---|---|---|
| Sequence-based reagent (*Inquilinitermes inquilinus*) | IinqOR and ORco sequences | *Johny et al., 2023* | | |
| Sequence-based reagent (*Neotermes cubanus*) | NcubOR and ORco sequences | *Johny et al., 2023* | | |
| Sequence-based reagent (*Prorhinotermes simplex*) | PsimOR and ORco sequences | *Johny et al., 2023* | | |

## Termites

Multiple laboratory colonies of *P. simplex*, originating from previous field collections in Cuba and Florida, are held in the Institute of Organic Chemistry and Biochemistry, Czech Academy of Sciences. Colonies are reared in glass vivaria at 27°C and 80% relative humidity in clusters of spruce wood slices.

The data reported here were collected from three mature Cuban colonies. The first one was used for antennal transcriptome sequencing and assembly followed by phylogenetic analysis, as described in *Johny et al., 2023*. The second one was used for RNA extraction for OR cloning, SSR, SEM, and HR-SEM. The third one was used for caste-specific head transcriptomes (head + antennae) of workers and soldiers, for caste-specific EAG recordings, and for SSR confirmation of the neocembrene-detecting sensillum. For all experiments with workers, fourth or fifth stage workers were selected as the most abundant developmental stages, recognized according to body size and head width.

## RNA extraction, OR cloning, and construct generation

Total RNA was extracted from 20 pairs of dissected *P. simplex* antennae using PureLink RNA Mini Kit (Invitrogen, Carlsbad, CA, USA) following the manufacturer's protocol and quantified using NanoDrop spectrophotometer (Thermo, Delaware, USA). From the total RNA, 2 µg was used to synthesize the cDNA using SuperScript IV Reverse Transcriptase (Invitrogen, Carlsbad, CA, USA) according to the manufacturer's instructions. The efficiency of cDNA synthesis was evaluated by amplification of ORCo. The list of primers is provided in Key Resource Table.

The full-length open reading frame of each selected PsimORs was PCR-amplified from the cDNA using the DreamTaq Green PCR Master Mix (Invitrogen, USA) and gene-specific primers. Amplification products were purified by QIAquick Gel Extraction Kit (QIAGEN, Germany), cloned into pCR8/GW/TOPO vector using the TOPO TA Cloning Kit (Invitrogen, USA), and transformed into OneShot TOP10 competent cells (Invitrogen, USA). Positive colonies were selected based on colony PCRs using primers GW1 and GW2, recombinant plasmids were isolated using the QIAprep 2.0 Spin Miniprep Columns (QIAGEN, Germany) and sequences were verified by Sanger sequencing (Eurofins Genomics, Germany).

The expression vector constructs were prepared using the Gateway LR recombination cloning technology (Invitrogen, USA) based on recombination of the phage-like attachment sites attL/R in pCR8/GW/TOPO with the bacteria-like attachment site attB in pUASg.attb vector (obtained from *Drosophila* Genomics Resource Center, Bloomington, USA). The resulting constructs pUASg.attB-PsimOR were purified using the QIAprep 2.0 Spin Miniprep Columns (QIAGEN, Germany) and insert sequences were verified by Sanger sequencing at Eurofins Genomics (Germany). All primers used for Sanger sequencing and colony PCR are listed in Key Resource Table.

## Fly lines

*D. melanogaster* lines used in the Empty Neuron system were kindly provided by Dr. Thomas O. Auer (from Richard Benton Lab, University of Lausanne, Switzerland). The wild-type $W^{1118}$ line, used as a control, was kindly provided by Prof. Michal Žurovec (Biology Centre, Czechia). All *D. melanogaster* lines were reared in an incubator which was set at 24 ± 2°C with relative humidity of 50 ± 5%. Flies

were fed with in-house prepared diet based on standard cornmeal food. The fly lines used are listed in Key Resource Table.

## Transgenic expression of termite ORs in *D. melanogaster* ab3A neuron

Selected PsimORs were expressed in the *D. melanogaster* Empty Neuron system for functional screening. Transgenic *D. melanogaster* UAS-PsimOR lines were generated by BestGene Inc (Chino Hills, CA, USA) by injecting pUASg.attB-PsimOR vectors into fly embryos expressing the integrase PhiC31 and carrying an attP landing site resulting in flies with genotype w−; +; UAS-PsimOR (w+)/+.

The recent CRISPR–Cas9-engineered empty neuron line Or22ab$^{-Gal4}$ (*Chahda et al., 2019*) was used as Δhalo genetic background for the expression of UAS-PsimOR in Dmel ab3 sensilla. The fly crossing scheme was adapted from *Gonzalez et al., 2016* with a modification at the F3 crossing. Final homozygote lines with UAS-PsimOR and Or22ab$^{-Gal4}$ were generated and used for the electrophysiological recordings. The full description of the crossing scheme is provided in *Figure 2—figure supplement 1*.

## Chemicals

For SSR measurements, we used a total of 67 chemicals organized into four panels. The initial screening Panel 1 contained 11 compounds biologically relevant to termites, that is components of termite pheromones and CHCs known from Neoisoptera and structurally related compounds. This panel was used for initial SSR screening of PsimOR9, 14, 30, and 31 in transgenic *D. melanogaster* and for SSR experiments with *P. simplex* workers. For detailed analysis of Psim OR14, three additional panels were used, consisting of 56 frequently occurring insect semiochemicals (e.g., terpenoids, fatty acid esters, fatty alcohols and aldehydes, etc.). Panel 1 compounds were diluted in *n*-hexane to 100 ng/µl, Panel 2–4 compounds were diluted in paraffin oil to $10^{-3}$ vol/vol. List of all compounds tested and their origin is listed in *Figure 3—source data 4*.

## Organic synthesis

For the purpose of SSR experiments, we synthesized (*Z*)-dodec-3-en-1-ol, (3*Z*,6*Z*)-dodeca-3,6-dien-1-ol, (3*Z*,6*Z*,8*E*)-dodecatrien-1-ol, and dodec-3-yn-1-ol, and included these compounds into Panel 1. The *de novo* organic synthesis of these fatty alcohols is described in Appendix 1.

## Electrophysiology

SSR recordings on *Drosophila* ab3 sensillum were performed as described previously (*Benton and Dahanukar, 2023*; *Olsson and Hansson, 2013*). We used 2- to 4-day-old flies for one recording each to avoid neuronal adaptations from multiple stimulations. To expose more ab3 sensilla, the fly preparation was done with arista down (*Keesey et al., 2022*).

In termites, the olfactory sensilla situated on the last antennal flagellomere of workers were targeted for SSR, since their number increases toward the distal end of termite antennae, the last segment being significantly more populated by olfactory sensilla than any other segment (*Castillo et al., 2021*). The grounding electrode was carefully inserted into the clypeus, and the antenna was fixed on a microscope slide using a glass electrode. To avoid the antennal movement, the microscope slide was covered with double-sided tape and the three distal antennal segments were attached to the slide.

The sensilla were observed under the Nikon FN1 eclipse microscope at ×60 magnification. For all electrophysiological measurements, the recording electrode was brought into contact with the base of the sensillum using a Kleindiek Nanotechnik MM3A micromanipulator connected to a cubic micromanipulator device. Using the Syntech stimulus delivery system (CS55 model, Syntech, Germany), the odorant stimulus was administered as a 0.3-s pulse by inserting the tip of the glass Pasteur pipette through an opening into the delivery tube (situated 4 cm from the tube outlet) carrying a purified air stream (0.4 l/min). The tube outlet was placed approximately 1 cm from the antenna. The experiments were conducted at 25–26°C.

From each diluted odorant (100 ng/µl), 10 µl were pipetted on 1-cm diameter filter paper disk placed in a glass Pasteur pipette in the screening experiment, while the doses ranging from 0.01 to 500 ng of neocembrene per filter paper were used in the dose–response experiments.

The signal was amplified and digitally converted using Syntech IDAC-4. The neuronal cells were sorted based on their amplitude, and the spikes were counted using the AutoSpike v3.9 software

(Syntech Ockenfels, Germany). Δ spike was calculated by subtracting the number of spikes during 1s post-stimulation from the number of spikes generated 1 s before the stimulation. In dose–response experiments and measurements on termite antennae, the counting periods were 0.5 s. Δ spike values were corrected by subtracting the response generated by the solvent and converted into Δ spike/s (*Benton and Dahanukar, 2023*; *Olsson and Hansson, 2013*). The receptor lifetime sparseness value was calculated according to *Chang et al., 2023*.

EAG experiment addressing the caste specificity of antennal responses to neocembrene was performed with 15 workers and 15 soldiers; each individual was only used for one stimulation series consisting of air–hexane–neocembrene (10 ng)–hexane–air. The brain and antennal tip were placed between two Ag/AgCl electrodes containing Ringer's solution and connected to a high impedance ($10^{14}$ Ω) amplifier (Syntech, Buchenbach, Germany). The antennal preparation was placed into a stream of cleaned air (500 ml/min), into which the stimuli were injected from Pasteur pipettes containing a 1.5-cm$^2$ filter paper impregnated with 10 µl of the tested solution. Odor injections were controlled by a foot switch-operated Syntech stimulus controller and maximal negative deflection was measured using Syntech EagPro software. Pasteur pipettes containing odorant stimuli were changed after three stimulations. Air responses were used for data normalization, the responses $\log_2$-transformed to reduce heteroscedasticity and comply with assumptions for parametric testing (Bartlett test for equal variances, and Shapiro–Wilk normality test), and then compared between workers and soldiers using Student's *t*-test.

## Scanning electron microscopy

For SEM, 10 workers with intact antennae were cold-anesthetized and decapitated with micro-scissors. Heads were desiccated in increasing ethanol concentrations (60, 80, 90, and 96%, each for 2 hr) followed by 12 hr in acetone. Heads were then attached to aluminum holders for microscopy using adhesive tape and differentially oriented to allow axial, dorsal, ventral, and lateral views. The samples were gold-coated for regular SEM (4 nm gold layer) and HR-SEM (2 nm) using sputter coater Bal-Tec SCD 050. Last antennal segments were inspected and photographed under scanning electron microscope JEOL 6380 LV (SEM). The surface of particular sensilla was studied using high-resolution field emission scanning electron microscope JSM-IT800 (HR-SEM) and Olympus Soft Imaging Solution software. Working distance for all samples was 4.0–4.1 mm and accelerating voltage 2.0 kV.

## Bioinformatics

For phylogenetic reconstruction of termite ORs, we used 182 OR protein sequences originating from five termite species, that is *Neotermes cubanus*, *P. simplex*, *Inquilinitermes inquilinus* (*Johny et al., 2023*), *Zootermopsis nevadensis* (*Terrapon et al., 2014*), and *Cryptotermes secundus* (*Harrison et al., 2018*), and the bristletail *L. saccharina* (*Thoma et al., 2019*) as a basal insect outgroup. For all species, the ORCo sequence was included. Sequences were aligned by means of the MUSCLE algorithm and used for reconstructing the phylogenetic tree with the IQ-TREE maximum likelihood algorithm (*Nguyen et al., 2015*) using the JTT+F+R8 substitution model and 10,000 ultrafast boot-strap replicates.

The gene structures of *PsimORs* were characterized by local alignment of full-length transcript sequences from *Johny et al., 2023* to our in-house genome assembly using BLAST (for details on genome assembly see *Koubová et al., 2021*) and confirmed with genomic mapping of the RNAseq data from *P. simplex* antennae available under accession SRX17749141 in NCBI SRA archives using STAR aligner v2.7.10b (*Dobin et al., 2013*). The mapping results were further used for abundance estimations of all ORs and ORCo in antennal transcriptome reported in *Johny et al., 2023*. Read counts were obtained with the featureCounts tool from the Subread package (https://subread.source-forge.net/) and normalized according to the FPKM (Fragments Per Kilobase Million) method.

Differential OR expression analysis in *P. simplex* soldier and worker heads (including antennae) was performed using the RNAseq data from our previous sequencing project and available as SRA archives under accessions SRX18952230–32 and SRX18952237–39. The data was obtained from sequencing of three biological replicates of each caste. Read counts obtained using STAR mapping and featureCounts estimations were statistically evaluated using the DESeq2 Bioconductor package in R and edgeR.

PsimOR14 secondary structure was predicted using online tools Jpred 4.0.0 (http://www.compbio.dundee.ac.uk/jpred) and TMHMM2.0 (https://services.healthtech.dtu.dk/services/TMHMM-2.0), schematic model was generated using Protter (https://wlab.ethz.ch/protter).

## Protein modeling

PsimOR14 structure was modeled in its monomeric membrane-free form using AlphaFold2 (*Jumper et al., 2021*; *Mirdita et al., 2022*). The best model was refined by MD relaxation, employing GROMACS 2021.3 and CHARMM36m (*Abraham et al., 2015*; *Huang et al., 2017*). After solvation and neutralization by Na$^+$ ions in TIP3P CHARMM water in a 1.5-nm padded box, temperature and pressure equilibration followed. Six different simulations with differing starting velocities were produced, followed by 150 ns periodic simulated annealing independently for each replica (0.5 ns at 300 K, then 0.5 ns at 320 K, repeating). The lowest potential energy structure was chosen. Neocembrene, geranylgeraniol, and (+)-limonene structures were sourced from PubChem (February 13, 2023) and parametrized using CgenFF 4.6, employing CHARMM-GUI for the conversion (*Jo et al., 2008*; *Kim et al., 2023*; *Vanommeslaeghe et al., 2010*). The binding site was predicted based on DEET-binding region of *Mh*OR5 from *M. hrabei* (7LIG) (*Del Mármol et al., 2021*). Using the DockThor webserver, ligands were docked with all bonds treated as rotatable, centered around the expected binding site with maximized box size (40 units). The best binder was then selected for each complex (*Santos et al., 2020*).

## MM/PBSA simulations

Complex topologies were built in GROMACS. Simulations included nine replicas for liganded (14 μs each) and six for unliganded PsimOR14 (6.9 μs), with different starting velocities. Convergence was assessed by ligand backbone RMSD distributions. Replicas were concatenated, split into 1.5 ns frames, and analyzed with MM/PBSA (ff19SB+GAFF2, linearized PB with diel = 2, SASA for apolar contributions, optimized CHARMM radii) using gmx_MMPBSA v1.6.3 and AmberTools 20 (*Case et al., 2023*; *Guedes et al., 2021*; *Tian et al., 2020*; *Wang et al., 2004*). For each liganded PsimOR14 (neocembrene, geranylgeraniol, and limonene), nine replica trajectories were separately analyzed, along with apoform PsimOR14 trajectories (six replicas). All trajectories were PBC-corrected, cleaned, and fitted to the first frame of the PsimOR14 trajectory.

Atom positions were marked in 3D space per frame, and the convex hull algorithm approximated the volume explored by each atom. This compared dynamic behavior between trajectories, with unliganded PsimOR14 as the baseline. The average total volume explored by atoms was compared.

## Acknowledgements

This study was supported by the Czech Science Foundation (No. 20-17194 S) and Institute of Organic Chemistry and Biochemistry, CAS (RVO: 61388963). We thank M Hyliš and the Viničná Microscopy Core Facility (VMCF of the Faculty of Science, Charles University) supported by the MEYS CR (LM2023050 Czech-BioImaging) for their support and assistance with microscopy techniques. We also thank Martina Hajdušková (https://www.biographix.cz/) for insect drawings used in *Fig. 1AFigure 1A*.

## Additional information

### Funding

| Funder | Grant reference number | Author |
| --- | --- | --- |
| Czech Science Foundation | 20-17194S | Souleymane Diallo<br>Kateřina Kašparová<br>Jibin Johny<br>Jan Křivánek<br>Pavlína Kyjaková<br>Ales Machara<br>Ondřej Lukšan<br>Ewald Grosse-Wilde<br>Robert Hanus |

| Funder | Grant reference number | Author |
|---|---|---|
| Institute of Organic Chemistry and Biochemistry, Czech Academy of Sciences | 61388963 | Souleymane Diallo<br>Kateřina Kašparová<br>Josef Šulc<br>Jan Křivánek<br>Pavlína Kyjaková<br>Jiří Vondrášek<br>Aleš Machara<br>Ondřej Lukšan<br>Robert Hanus |
| MEYS | LM2023050 | Jana Nebesářová |

The funders had no role in study design, data collection, and interpretation, or the decision to submit the work for publication.

## Author contributions

Souleymane Diallo, Data curation, Formal analysis, Validation, Investigation, Writing – original draft; Kateřina Kašparová, Josef Šulc, Investigation; Jibin Johny, Resources, Investigation, Methodology; Jan Křivánek, Pavlína Kyjaková, Resources, Investigation; Jana Nebesářová, Supervision, Investigation; David Sillam-Dussès, Resources, Methodology; Jiří Vondrášek, Formal analysis, Supervision, Investigation; Aleš Machara, Resources, Validation, Investigation, Methodology; Ondřej Lukšan, Conceptualization, Resources, Data curation, Formal analysis, Supervision, Validation, Investigation, Methodology; Ewald Grosse-Wilde, Conceptualization, Resources, Data curation, Formal analysis, Supervision, Funding acquisition, Validation, Investigation, Project administration; Robert Hanus, Conceptualization, Resources, Formal analysis, Supervision, Funding acquisition, Investigation, Writing – original draft, Project administration, Writing - review and editing

## Author ORCIDs

Jibin Johny ⓘ https://orcid.org/0000-0002-2265-5046
Pavlína Kyjaková ⓘ https://orcid.org/0000-0003-0081-3717
Jiří Vondrášek ⓘ https://orcid.org/0000-0002-6066-973X
Robert Hanus ⓘ https://orcid.org/0000-0002-7054-1975

Reviewer #1 (Public review): https://doi.org/10.7554/eLife.101814.3.sa1
Reviewer #2 (Public review): https://doi.org/10.7554/eLife.101814.3.sa2
Reviewer #3 (Public review): https://doi.org/10.7554/eLife.101814.3.sa3
Author response https://doi.org/10.7554/eLife.101814.3.sa4

# Additional files

## Supplementary files
MDAR checklist

## Data availability

Nucleotide and protein sequences of termite ORs used for phylogenetic reconstruction and functional characterizations were published in *Johny et al., 2023* and the raw sequencing data was previously deposited in NCBI SRA archive as PRJNA885453 BioProject. Transcript abundances of *P. simplex* ORs were also inferred from the antennal transcriptome data, available at NCBI under SRX17749141. Sequences of PsimOR14 studied in detail in the present paper are listed in *Figure 6—source data 1* and deposited at NCBI as OR921181 entry. Origin of the *P. simplex* draft genome assembly is reported in *Koubová et al., 2021*. Differential expression of *P. simplex* ORs in workers and soldiers was studied using caste-specific head transcriptomes available at NCBI as SRA archives under accessions SRX18952230-32 and SRX18952237-39. All data generated in this article are reported in the source data files related to individual figures.

The following datasets were generated:

| Author(s) | Year | Dataset title | Dataset URL | Database and Identifier |
|---|---|---|---|---|
| Lukšan O, Hanus R | 2023 | RNA-Seq of Prorhinotermes simplex: antennae | https://www.ncbi.nlm.nih.gov/sra/?term=SRX17749141 | NCBI Sequence Read Archive, SRX17749141 |
| Diallo S, Kasparova K, Hanus R, Sillam-Dusses D, Johny J, Pflegerova J, Krivanek J | 2023 | Prorhinotermes simplex neocembren-detecting odorant receptor (OR14) gene, complete cds | https://www.ncbi.nlm.nih.gov/nuccore/OR921181.1 | NCBI Nucleotide, OR921181 |

The following previously published datasets were used:

| Author(s) | Year | Dataset title | Dataset URL | Database and Identifier |
|---|---|---|---|---|
| Lukšan O, Hanus R | 2022 | Antennal transcriptome of three termite species | https://www.ncbi.nlm.nih.gov/bioproject/?term=PRJNA885453 | NCBI BioProject, PRJNA885453 |
| Lukšan O, Hanus R | 2023 | RNAseq of Prorhinotermes simplex: Worker head | https://www.ncbi.nlm.nih.gov/sra/?term=SRX18952239 | NCBI Sequence Read Archive, SRX18952239 |
| Lukšan O, Hanus R | 2023 | RNAseq of Prorhinotermes simplex: Worker head | https://www.ncbi.nlm.nih.gov/sra/?term=SRX18952238 | NCBI Sequence Read Archive, SRX18952238 |
| Lukšan O, Hanus R | 2023 | RNAseq of Prorhinotermes simplex: Worker head | https://www.ncbi.nlm.nih.gov/sra/?term=SRX18952237 | NCBI Sequence Read Archive, SRX18952237 |
| Lukšan O, Hanus R | 2023 | RNAseq of Prorhinotermes simplex: Soldier head | https://www.ncbi.nlm.nih.gov/sra/?term=SRX18952232 | NCBI Sequence Read Archive, SRX18952232 |
| Lukšan O, Hanus R | 2023 | RNAseq of Prorhinotermes simplex: Soldier head | https://www.ncbi.nlm.nih.gov/sra/?term=SRX18952231 | NCBI Sequence Read Archive, SRX18952231 |
| Lukšan O, Hanus R | 2023 | RNAseq of Prorhinotermes simplex: Soldier head | https://www.ncbi.nlm.nih.gov/sra/?term=SRX18952230 | NCBI Sequence Read Archive, SRX18952230 |

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

## Appendix 1

### Organic synthesis of (3Z)-dodec-3-en-1-ol, dodec-3-yn-1-ol, (3Z,6Z)-dodeca-3,6-dien-1-ol, and (3Z,6Z,8E)-dodeca-3,6,8-trien-1-ol

Unless noted otherwise, all reactions were carried out under argon in oven-dried glassware. Solvents were distilled from drying agents as indicated and transferred under nitrogen: Tetrahydrofuran (Na/benzophenone) and toluene (Na/benzophenone). All starting materials were used as purchased (Sigma-Aldrich, Combi-Blocks), unless otherwise indicated. Chromatography was performed using Fluka silica gel 60 (0.040–0.063 mm). For Thin Layer Chromatography analysis, F254-coated aluminum sheets were used. The spots were detected both in UV and by the solution of $Ce(SO_4)_2 \cdot 4H_2O$ (1%) and $H_3P(Mo_3O_{10})_4$ (2%) in 10% sulfuric acid. $^1$H- and $^{13}$C NMR spectra were recorded at 400 and 100 MHz, respectively, with a Bruker 400-MHz instrument at 25°C (the solvents are indicated in parentheses). Chemical shifts are reported in ppm relative to TMS. The residual solvent signals in the $^1$H- and $^{13}$C NMR spectra were used as an internal reference (CDCl$_3$: $\delta$ = 7.26 for $^1$H and $\delta$ = 77.23 for $^{13}$C). The compounds were analyzed using the gas chromatograph TRACE 1310 (Thermo Fisher Scientific, Waltham, MA, USA), equipped with a nonpolar Zebron ZB-5MS column (30 m × 0.25 mm × 0.25 μm film; Phenomenex, Torrance, CA, USA) connected to a Thermo Fisher Scientific ISQ LT mass-selective detector (70 eV ionization voltage, source temperature 200°C, transfer line heated to 260°C). The column temperature was held at 50°C for 1 min, gradually increased to 320°C at 8°C/min and then held at 320°C for 20 min. Helium was used as a carrier gas at a flow of 1.2 ml/min. Split/splitless port was heated to 200°C, and samples were injected in splitless mode with a purge time of 1 min.

**Appendix 1—figure 1.** Synthesis of 4 and 5.

### 2-(But-3-yn-1-yloxy)tetra-2H-hydropyran (*Allegretti and Ferreira, 2011*) (1)

To a solution of 3-butyn-1-ol (3.0 g; 42.79 mmol) in DCM (100 ml) at 0°C was added 3,4-dihydro-2H-pyran (3.78 g; 44.93 mmol) followed by addition of p-toluenesulfonic acid monohydrate (0.080 g; 0.427 mmol). After stirring at 0°C for 5 min, the reaction was warmed to room temperature and stirred for 90 min. The reaction was quenched with a saturated solution of NaHCO$_3$ (20 ml), and the layers were separated. The aqueous layer was extracted with DCM (50 ml), and the combined organic layers were dried over MgSO$_4$ and concentrated in vacuo. The residue was purified by a column chromatography (silica gel; cyclohexane/EtOAc 4:1) to give **4** as colorless oil (0.53 g, 64%). $^1$H NMR (401 MHz, CDCl$_3$) δ 4.67 (dd, J = 4.2, 2.9 Hz, 1H), 3.97–3.81 (m, 2H), 3.65–3.49 (m,

2H), 2.52 (td, *J* = 7.0, 2.7 Hz, 2H), 2.00 (t, *J* = 2.7 Hz, 1H), 1.93–1.78 (m, 1H), 1.78–1.69 (m, 1H), 1.69–1.46 (m, 4H). EI-MS (70 eV): *m/z* (%): 153 (1.4), 125 (2), 99 (9), 85 (100), 79 (9), 67 (21), 53 (33), 41 (22).

## 2-(Dodec-3-yn-1-yloxy)tetrahydro-2*H*-pyran (2)

To a solution of acetylene **1** (1.0 g; 6.48 mmol) in THF (10 ml) at –40°C under argon atmosphere was added *n*-BuLi (2.6 ml, 2.5 M in hexanes, 6.5 mmol), followed with addition of HMPA (1 ml) and the resulting solution was stirred for 40 min at –40°C. Then a solution of octyl bromide (1.25 g; 6.48 mmol) in THF (2.00 ml) was added rapidly. The resulting mixture was stirred at –40°C for 60 min, and then warmed to room temperature and stirred for 2 hr. The reaction mixture was quenched by addition of water (10 ml) and was diluted with EtOAc (20 ml). The layers were separated, and the aqueous layer was extracted with EtOAc. The combined organic layers were washed with water and brine and concentrated under reduced pressure. The residue was purified by a column chromatography (silica gel; cyclohexane/EtOAc 9:1) to give **2** as colorless oil (1.12 g, 86%). $^1$H NMR (401 MHz, CDCl$_3$) δ 4.67 (dd, *J* = 4.2, 2.9 Hz, 1H), 3.91 (ddd, *J* = 11.2, 8.1, 3.3 Hz, 1H), 3.82 (dt, *J* = 9.7, 7.2 Hz, 1H), 3.61–3.47 (m, 2H), 2.48 (tt, *J* = 7.2, 2.4 Hz, 2H), 2.16 (tt, *J* = 7.1, 2.4 Hz, 2H), 1.94–1.79 (m, 1H), 1.74 (tdd, *J* = 9.2, 3.9, 2.9 Hz, 1H), 1.67–1.43 (m, 6H), 1.43–1.24 (m, 10H), 0.95–0.86 (m, 3H). $^{13}$C NMR (101 MHz, CDCl$_3$) δ 98.7, 81.4, 66.3, 62.2, 31.9, 30.6, 29.2, 29.2, 29.0, 28.9, 25.5, 20.3, 19.4, 18.8, 14.1. EI-MS (70 eV): *m/z* (%): 265 (0.002), 211 (0.4), 195 (0.7), 153 (2), 115 (4), 101 (7), 85 (100), 81 (8), 67 (17), 55 (13), 41 (12).

## 2-[((*Z*)-Dodec-3-enyl)oxy]-tetrahydro-2*H*-pyran (3)

To a solution of borane-dimethylsulfide complex (0.44 g; 5.78 mmol) in THF (10 ml) at –10°C under argon atmosphere cyclohexane (0.95 g; 11.57 mmol) was added. After the addition was complete, the mixture was slowly warmed to room temperature over 3 hr, yielding a white slurry. The mixture was cooled to –20°C and a solution of substrate 2 (0.77 g; 2.89 mmol) in THF (3 ml) was added dropwise. The reaction mixture was slowly warmed to 0°C over 3 hr and stirred at 0°C for a further 3 hr. Then acetic acid (1.40 g; 23.1 mmol) was then added dropwise, and the reaction mixture was stirred overnight. The reaction mixture was cooled to 0°C, and a solution of NaOH (1.9 g) in water (8 ml) was added over 10 min, followed by dropwise addition of hydrogen peroxide (0.7 ml, 30% solution in water). The stirred mixture was warmed to room temperature and poured into water. Crude product was extracted with cyclohexane and then concentrated under reduced pressure. The residue was purified by column chromatography (silica gel; cyclohexane/EtOAc 9:1) to provide **3** as colorless oil (0.71 g, 91%). $^1$H NMR (401 MHz, CDCl$_3$) δ 5.56–5.28 (m, 2H), 4.63 (dd, *J* = 4.4, 2.7 Hz, 1H), 3.97–3.85 (m, 1H), 3.75 (dt, *J* = 9.5, 7.2 Hz, 1H), 3.61–3.48 (m, 1H), 3.48–3.36 (m, 1H), 2.46–2.32 (m, 2H), 2.06 (td, *J* = 7.1, 1.3 Hz, 2H), 1.86 (ddt, *J* = 10.7, 7.8, 5.3 Hz, 1H), 1.80–1.70 (m, 1H), 1.69–1.50 (m, 4H), 1.38–1.28 (m, 10H), 0.94–0.87 (m, 3H). $^{13}$C NMR (101 MHz, CDCl$_3$) δ 132.1, 125.4, 98.8, 67.1, 62.3, 31.9, 30.7, 29.7, 29.5, 29.4, 29.2, 28.90, 27.4, 22.7, 19.6, 14.1. EI-MS (70 eV): *m/z* (%): 268 (0.1), 166 (3), 115 (2), 101 (15), 85 (100), 67 (13), 55 (11), 41 (8).

## (*Z*)-Dodec-3-en-1-ol (4)

A mixture of *Z*-alkene **3** (0.71 g; 2.64 mmol), and *p*-toluenesulfonic acid monohydrate (0.025 g; 0.13 mmol) in methanol (2 ml) was stirred at room temperature for 3 hr. Then the reaction mixture was poured into an ice-cold solution of sodium bicarbonate and was extracted with diethyl ether (2 × 10 ml). The combined organic layers were washed with brine (5 ml), dried over MgSO$_4$, and evaporated. The residue was purified by column chromatography (silica gel; cyclohexane/EtOAc 9:1) to provide **4** as colorless oil (0.43 g, 87%). $^1$H NMR (401 MHz, CDCl$_3$) δ 5.66–5.48 (m, 1H), 5.46–5.31 (m, 1H), 3.67 (t, *J* = 6.5 Hz, 2H), 2.40–2.32 (m, 2H), 2.13–1.98 (m, 2H), 1.43–1.21 (m, 12H), 0.97–0.86 (m, 3H). $^{13}$C NMR (101 MHz, CDCl$_3$) δ 133.6, 124.9, 62.4, 31.9, 30.8, 29.7, 29.5, 29.3, 29.3, 27.4, 22.7, 14.1. EI-MS (70 eV): *m/z* (%): 184 (0.1), 166 (5), 138 (6), 124 (8), 110 (18), 96 (39), 81 (79), 68 (100).

## Dodec-3-yn-1-ol (5)

A mixture of THP-protected alkyne **2** (0.30 g; 1.12 mmol), and *p*-toluenesulfonic acid monohydrate (0.04 g; 0.022 mmol) in methanol (5 ml) was stirred at room temperature for 3 hr. Then the reaction mixture was poured into an ice-cold solution of sodium bicarbonate and was extracted with diethyl ether (2 × 10 ml). The combined organic layers were washed with brine (5 ml), dried over MgSO$_4$, and evaporated. The residue was purified by column chromatography (silica gel; cyclohexane/EtOAc 4:1) to provide **5** as colorless oil (0.20 g, 98%). $^1$H NMR (401 MHz, CDCl$_3$) δ 3.70 (t, *J* = 6.2 Hz, 2H), 2.46 (tt, *J* = 6.2, 2.4 Hz, 2H), 2.18 (tt, *J* = 7.2, 2.4 Hz, 2H), 1.57–1.46 (m, 2H), 1.39–1.22 (m, 8H),

0.94–0.86 (m, 3H). $^{13}$C NMR (101 MHz, CDCl$_3$) δ 82.9, 76.2, 61.4, 31.8, 29.2, 29.1, 29.0, 28.9, 23.2, 22.7, 18.8, 14.1.

**Appendix 1—figure 2.** Synthesis of 8.

## 1-Bromooct-2-yne (6)

A mixture of 2-octynol (2.0 g; 15.87 mmol), triphenylphosphine (4.57 g; 17.46 mmol), and tetrabromomethane (6.30 g; 19.04 mmol) in DCM (40 ml) was stirred at 0°C for 2 hr and then it was diluted with cyclohexane and filtered. The filtrate was evaporated, and a column chromatography (silica gel; eluent cyclohexane) of the residue afforded 3.70 g (90%) of bromide **5** as a pale oil. $^1$H NMR (401 MHz, CDCl$_3$) δ 3.95 (t, J = 2.4 Hz, 2H), 2.25 (ddt, J = 7.2, 4.7, 2.4 Hz, 2H), 1.59–1.46 (m, 2H), 1.44–1.27 (m, 4H), 0.96–0.88 (m, 3H). $^1$H NMR data match published spectrum (*Sigurjónsson and Haraldsson, 2024*).

## Dodeca-3,6-diyn-1-ol (*Liu et al., 2018*) (7)

To a stirred solution of but-3-yn-1-ol (0.90 g; 12.76 mmol) in anhydrous DMF (30 ml) under an argon atmosphere at room temperature was added cesium carbonate (3.45 g; 10.63 mmol), sodium iodide (1.60 g; 10.63 mmol), and copper(I) iodide (2.02 g; 10.63 mmol). The reaction mixture was stirred for 30 min. Then a solution of 1-bromooct-2-yne (2.0 g; 10.63 mmol) in anhydrous DMF (10 ml) was added dropwise and stirring was continued for another 2 hr. The reaction mixture was quenched by addition of saturated aqueous NH$_4$Cl and extracted with Et$_2$O. The combined organic layers were washed with water, brine, and concentrated under reduced pressure. The residue was purified by a column chromatography (silica gel; cyclohexane/EtOAc 4:1) to provide diynol **7** as colorless oil (0.90 g, 45%). $^1$H NMR (401 MHz, CDCl$_3$) δ 3.73 (t, J = 6.2 Hz, 2H), 3.16 (t, J = 2.4 Hz, 2H), 2.47 (tt, J = 6.2, 2.4 Hz, 2H), 2.17 (tt, J = 7.2, 2.4 Hz, 2H), 1.56–1.46 (m, 2H), 1.43–1.22 (m, 4H), 0.98–0.85 (m, 3H). EI-MS (70 eV): m/z (%): 178 (0.2), 163 (0.7), 149 (2), 135 (4), 121 (14), 117 (20), 105 (37), 91 (100), 79 (41).

## (3Z,6Z)-Dodeca-3,6-dien-1-ol (*Liu et al., 2018*) (8)

To a stirred suspension of Ni(OAc)$_2$·H$_2$O (0.81 g; 3.25 mmol) in EtOH (10 ml) under an argon atmosphere at room temperature was added NaBH$_4$ (0.12 g; 3.25 mmol). Then the flask was filled with hydrogen, and a balloon with hydrogen gas was attached to the reaction flask for the rest of the experiment. Ethylenediamine (0.78 g; 13.02 mmol) was added to the reaction mixture after 30 min and then the reaction mixture was stirred for another 30 min. Then a solution of diynol **7** (0.58 g; 3.25 mmol) in EtOH (2 ml) was added and the mixture was stirred for another 2 hr. The reaction mixture was diluted with water and EtOAc (5+5 ml), and the P-2 catalyst was filtered through a pad of Celite. The filtrate was concentrated under reduced pressure and partitioned between water and Et$_2$O. The organic layer was separated, the aqueous phase was extracted with Et$_2$O, and the combined organic layers were washed with brine and concentrated under reduced pressure. The residue was purified by a column chromatography (silica gel; cyclohexane/EtOAc 4:1) to provide dienol **8** as colorless oil (0.29 g, 49%). $^1$H NMR (401 MHz, CDCl$_3$) δ 5.65–5.48 (m, 1H), 5.47–5.27 (m, 2H), 3.68 (t, J = 6.5 Hz, 2H), 2.89–2.75 (m, 2H), 2.44–2.32 (m, 2H), 2.13–1.99 (m, 2H), 1.47–1.22 (m, 6H), 0.98–0.86 (m, 3H). $^{13}$C NMR (101 MHz, CDCl$_3$) δ 131.6, 130.7, 127.4, 125.3, 62.3, 31.5, 30.8, 29.3, 27.3, 26.9, 25.8, 22.6, 14.1. EI-MS (70 eV): m/z (%): 182 (2), 164 (5), 138 (5), 135 (8), 121 (16), 107 (19), 93 (48), 79 (100), 67 (66).

**Appendix 1—figure 3.** Synthesis of **13**.

## (*E*)-1-Iodopent-1-ene (*Saran et al., 2007*) (9)

Diisobutylaluminum hydride in hexanes (1 M, 50 ml, 50 mmol) was added dropwise to a solution of pent-1-yne (3.7 g; 54 mmol) in hexanes cooled to –40°C. The resulting mixture was stirred for 30 min, then slowly warmed to room temperature over 3 hr, and was allowed to stir overnight. The mixture was warmed to ~50°C for 4 hr, then cooled to –40°C, and a solution of iodine (12.7 g; 50 mmol) in THF (20 ml) was added dropwise over 30 min. The resulting dark brown suspension was allowed to warm to room temperature with stirring overnight. The solution was cooled in an ice bath and slowly quenched with dropwise addition of ice-cold diluted sulfuric acid (5 ml of 96% acid in 60 ml of water) with vigorous stirring. After stirring for 30 min, the layers were separated, and the organic layer was washed with dilute $NaHSO_3$ solution and brine and dried over $MgSO_4$. The resulting solution was filtered through a short pad of silica gel, rinsing with pentane. The obtained solution was concentrated by rotary evaporation without heating (rotavap was set to 100 Torr pressure). This procedure gave 12.24 g of crude alkenyl iodide **9** that was used in the next step. EI-MS (70 eV): *m/z* (%): 155 (4), 141 (3), 127 (5), 73 (69), 55 (100).

## (*E*)-Oct-4-ene-2-yn-1-ol (*Saran et al., 2007*) (10)

To a mixture of bis(triphenylphosphine)palladium(II) dichloride (1.45 g; 2.0 mmol), copper(I) iodide (0.72 g; 3.8 mmol), and pyrrolidine (50 ml) under argon atmosphere the crude (*E*)-1-iodopent-1-ene (12.2 g) was added dropwise, followed by dropwise addition of propargyl alcohol (4.92 g; 87.8 mmol). The mixture was immersed into a water bath until the exothermic reaction ended. After cooling the mixture to room temperature, the mixture was stirred for 3 hr. Then the reaction mixture was poured into an ice-cold mixture of saturated aqueous $NH_4Cl$ and 2 M HCl (150 + 150 ml) and extracted several times with diethyl ether. The combined organic layers were washed with an aqueous solution of citric acid (10% solution), water, brine, and concentrated under reduced pressure. The residue was purified by a column chromatography (silica gel; eluent DCM) to provide alcohol **10** as colorless oil (2.22 g). $^1$H NMR (401 MHz, $CDCl_3$) δ 6.19 (dt, *J* = 15.9, 7.1 Hz, 1H), 5.57–5.47 (m, 1H), 4.48–4.36 (m, 2H), 2.11 (qd, *J* = 7.2, 1.6 Hz, 2H), 1.71–1.51 (m, 1H), 1.45 (q, *J* = 7.4 Hz, 2H), 0.93 (t, *J* = 7.4 Hz, 3H). EI-MS (70 eV): *m/z* (%): 124 (74), 109 (14), 95 (44), 81 (100), 67 (49), 53 (33).

## (*E*)-1-Bromooct-4-ene-2-yne (11)

To a stirred solution of alcohol **10** (2.22 g; 17.9 mmol) and triphenylphosphine (5.16 g; 19.7 mmol) in dry DCM (30 ml) under an argon atmosphere at 0°C was added solution of tetrabromomethane (7.12 g; 21.5 mmol) in dry DCM (10 ml). The reaction mixture was stirred at 0°C for 10 min and then an ice bath was removed, and the reaction mixture was allowed to stir for 3 hr. The reaction mixture was concentrated to half the volume and cyclohexane (30 ml) was added. The mixture was stirred for 30 min. Then the mixture was filtered through a pad of Celite, and the solvent was evaporated. A column chromatography (silica gel, eluent cyclohexane) of the residue gave 3.00 g (90%) of bromide **11** as colorless oil. $^1$H NMR (401 MHz, $CDCl_3$) δ 6.22 (dt, *J* = 15.9, 7.1 Hz, 1H), 5.58–5.47 (m, 1H), 4.08 (d, *J* = 2.3 Hz, 2H), 2.12 (qd, *J* = 7.2, 1.6 Hz, 2H), 1.51–1.37 (m, 4H), 0.93 (t, *J* = 7.4 Hz, 3H). $^{13}$C NMR (101 MHz, $CDCl_3$) δ 146.7, 108.8, 85.8, 82.5, 35.2, 26.9, 21.8, 15.8, 13.6. EI-MS (70 eV): *m/z* (%): 188 (21), 186 (23), 107 (100), 91 (55), 79 (49), 65 (66).

## (*E*)-Dodeca-8-en-3,6-diyn-1-ol (12)

To a stirred solution of but-3-yn-1-ol (1.58 g; 22.58 mmol) in anhydrous DMF (30 ml) under an argon atmosphere at room temperature was added cesium carbonate (6.11 g; 18.81 mmol), sodium iodide (2.82 g; 18.81 mmol), and copper(I) iodide (3.57 g; 18.81 mmol). The reaction mixture was stirred for 30 min. Then a solution of bromide **11** (3.50 g; 18.81 mmol) in anhydrous DMF (10 ml) was added dropwise and stirring was continued for another 2 hr. The reaction mixture was quenched by addition of saturated aqueous $NH_4Cl$ and extracted with $Et_2O$. The combined organic layers were washed with water, brine, and concentrated under reduced pressure. The residue was purified by a column chromatography (silica gel; cyclohexane/EtOAc 3:1) to provide alcohol **12** as colorless oil (2.0 g, 60%). $^1H$ NMR (401 MHz, $CDCl_3$) δ 6.14 (dt, *J* = 15.9, 7.1 Hz, 1H), 5.52–5.42 (m, 1H), 3.73 (t, *J* = 6.2 Hz, 2H), 3.33–3.26 (m, 2H), 2.52–2.42 (m, 2H), 2.14–2.03 (m, 2H), 1.51–1.36 (m, 2H), 0.92 (t, *J* = 7.4 Hz, 3H). $^{13}C$ NMR (101 MHz, $CDCl_3$) δ 144.7, 109.3, 81.9, 79.4, 77.2, 76.3, 61.1, 35.1, 23.2, 21.9, 13.6, 10.4. HRMS (ESI): *m/z* calcd for $C_{12}H_{16}ONa^+$: 199.1093 [M+Na]$^+$; found: 199.1094. EI-MS (70 eV): *m/z* (%): 176 (4), 161 (3), 147 (14), 128 (36), 115 (67), 103 (45), 91 (100), 77 (62).

## (3*Z*,6*Z*,8*E*)-Dodeca-3,6,8-trien-1-ol (13)

To a stirred suspension of $Ni(OAc)_2 \cdot H_2O$ (1.41 g; 5.68 mmol) in EtOH (8 ml) under an argon atmosphere at room temperature was added a suspension of $NaBH_4$ (0.21 g; 5.68 mmol) in EtOH (5 ml). Then the flask was filled with hydrogen, and a balloon with hydrogen gas was attached to the reaction flask for the rest of the experiment. Ethylenediamine (1.36 g; 22.7 mmol) was added to the reaction mixture after 30 min and then the reaction mixture was stirred for another 30 min. Then a solution of diynol **12** (1.0 g; 5.68 mmol) in EtOH (2 ml) was added and the mixture was stirred for another 2 hr. The reaction mixture was diluted with water and EtOAc (5 + 5 ml), and the P-2 catalyst was filtered through a pad of Celite. The filtrate was concentrated under reduced pressure and partitioned between water and $Et_2O$. The organic layer was separated, the aqueous phase was extracted with $Et_2O$, and the combined organic layers were washed with brine and concentrated under reduced pressure. The residue was purified by a column chromatography (silica gel; cyclohexane/EtOAc 4:1) to provide trienol **13** as colorless oil (0.40 g, 40%). $^1H$ NMR (401 MHz, $CDCl_3$) δ 6.42–6.25 (m, 1H), 6.12–5.93 (m, 1H), 5.72 (dt, *J* = 14.6, 7.0 Hz, 1H), 5.65–5.53 (m, 1H), 5.50–5.38 (m, 1H), 5.28 (dt, *J* = 10.6, 7.5 Hz, 1H), 3.69 (t, *J* = 6.5 Hz, 2H), 2.45–2.35 (m, 2H), 2.12 (qd, *J* = 7.5, 1.4 Hz, 2H), 1.51–1.44 (m, 2H), 0.94 (t, *J* = 7.3 Hz, 3H). $^{13}C$ NMR (101 MHz, $CDCl_3$) δ 135.5, 131.1, 129.1, 127.1, 125.7, 125.4, 62.3, 35.0, 30.9, 26.2, 22.5, 13.8. EI-MS (70 eV): *m/z* (%): 180 (13), 137 (8), 119 (20), 105 (41), 91 (100), 79 (75), 67 (42).

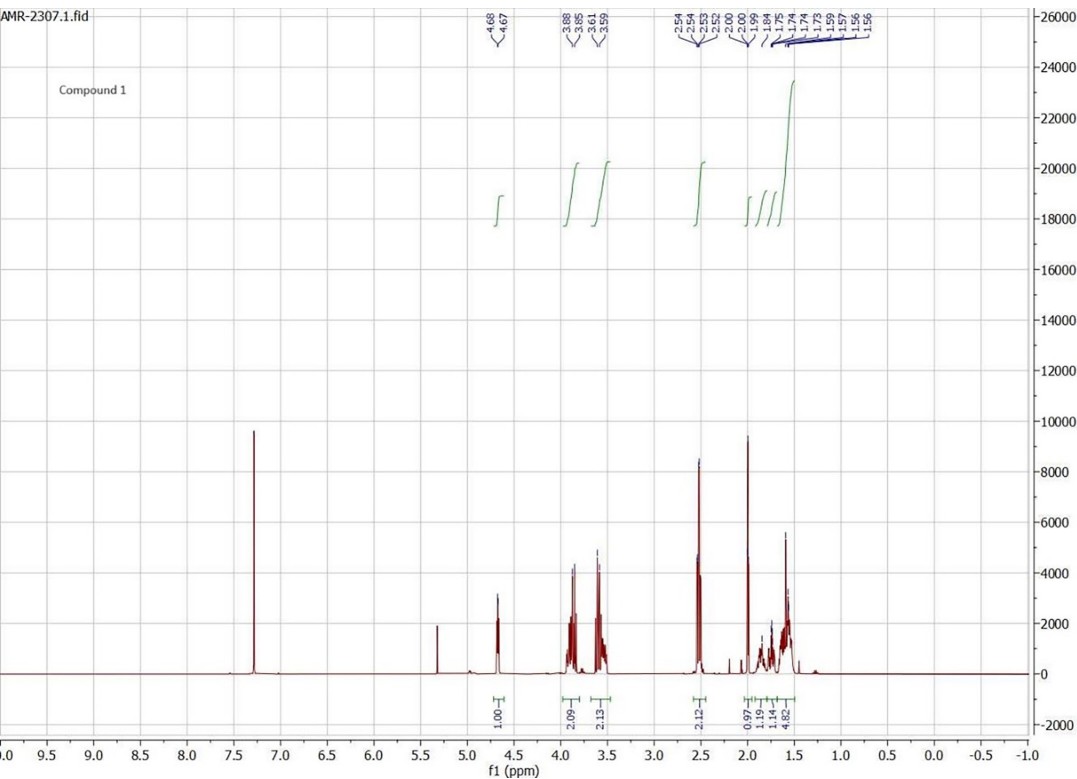

**Appendix 1—figure 4.** $^1$H NMR spectrum of compound 1.

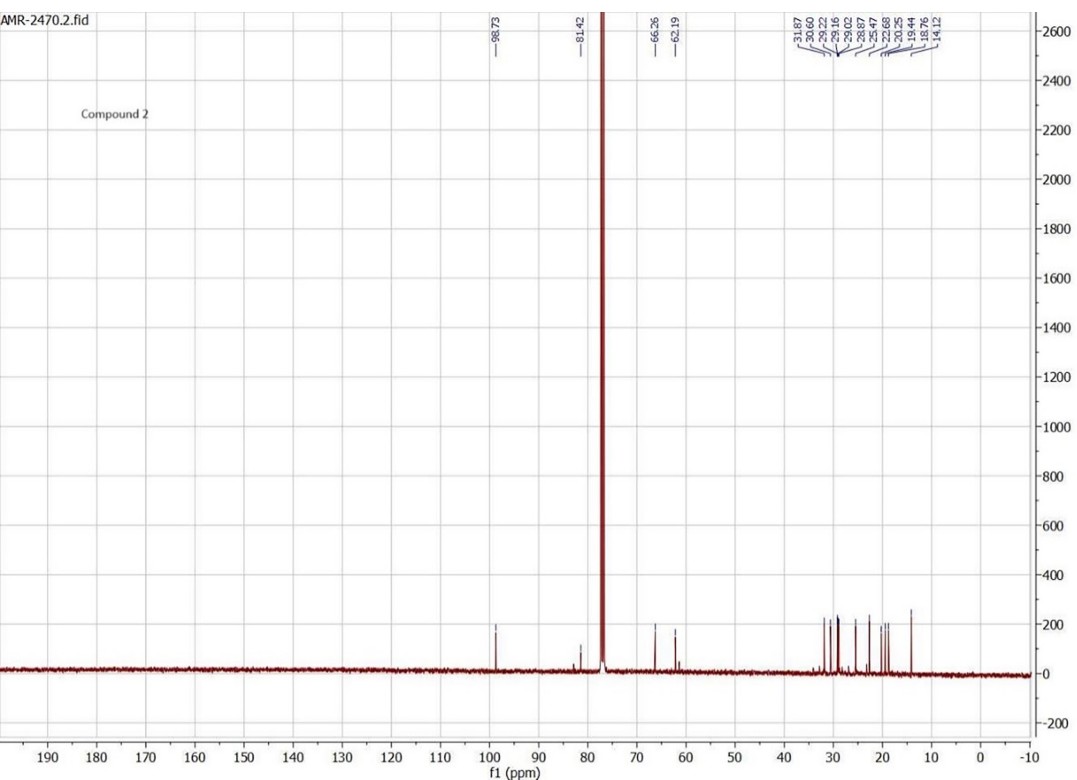

**Appendix 1—figure 5.** $^{13}$C NMR spectrum of compound 2.

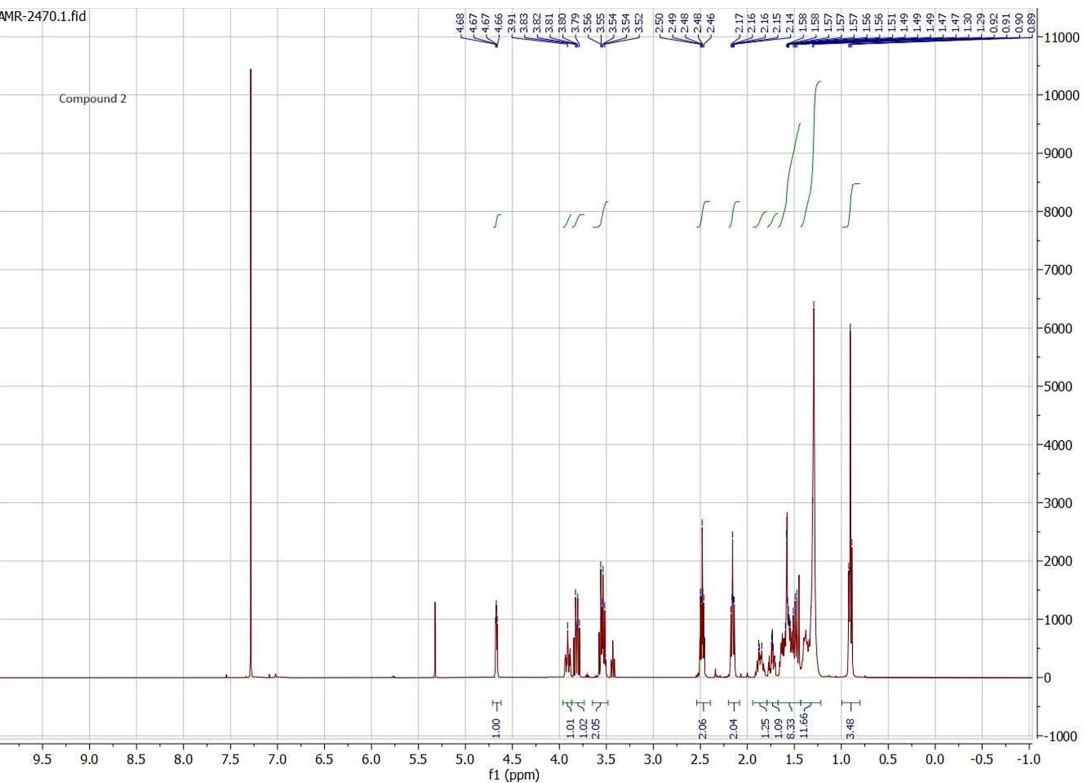

**Appendix 1—figure 6.** $^1$H NMR spectrum of compound **2**.

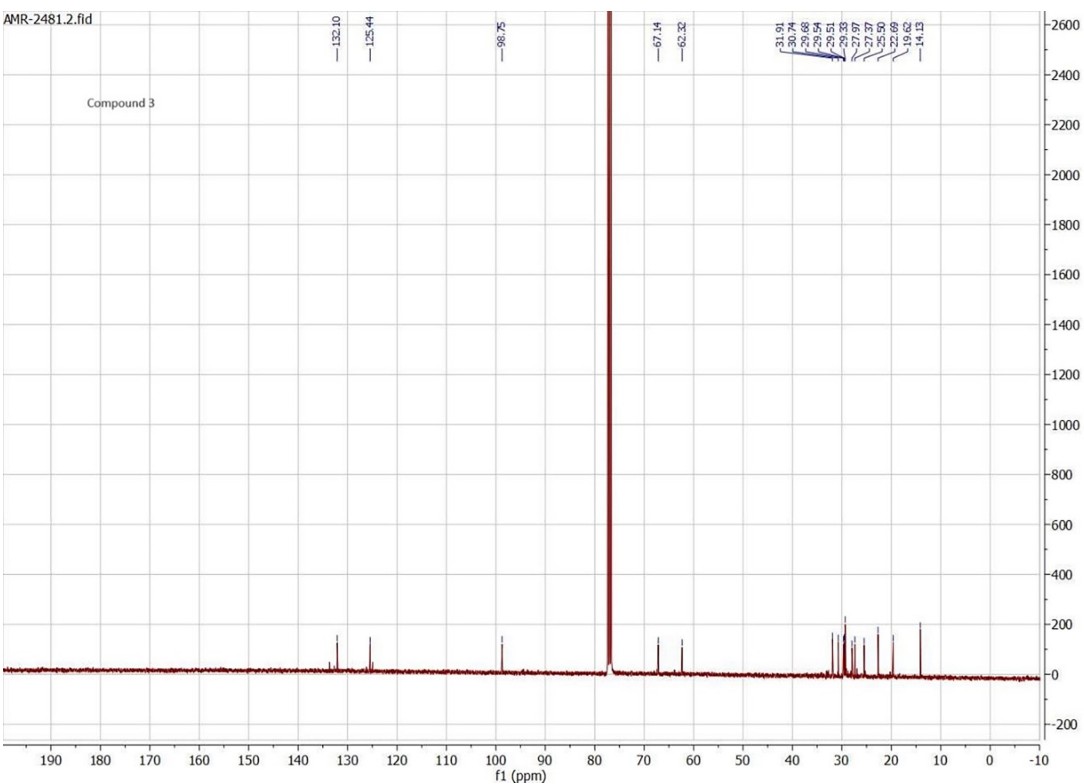

**Appendix 1—figure 7.** $^{13}$C NMR spectrum of compound **3**.

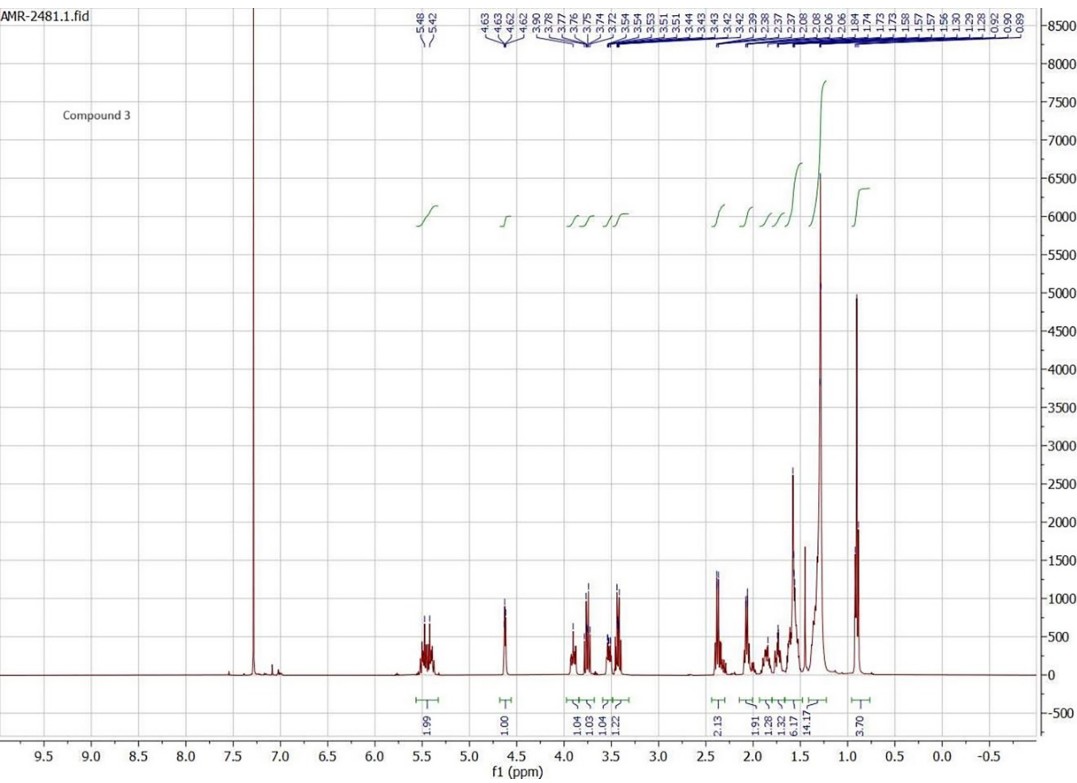

**Appendix 1—figure 8.** ¹H NMR spectrum of compound **3**.

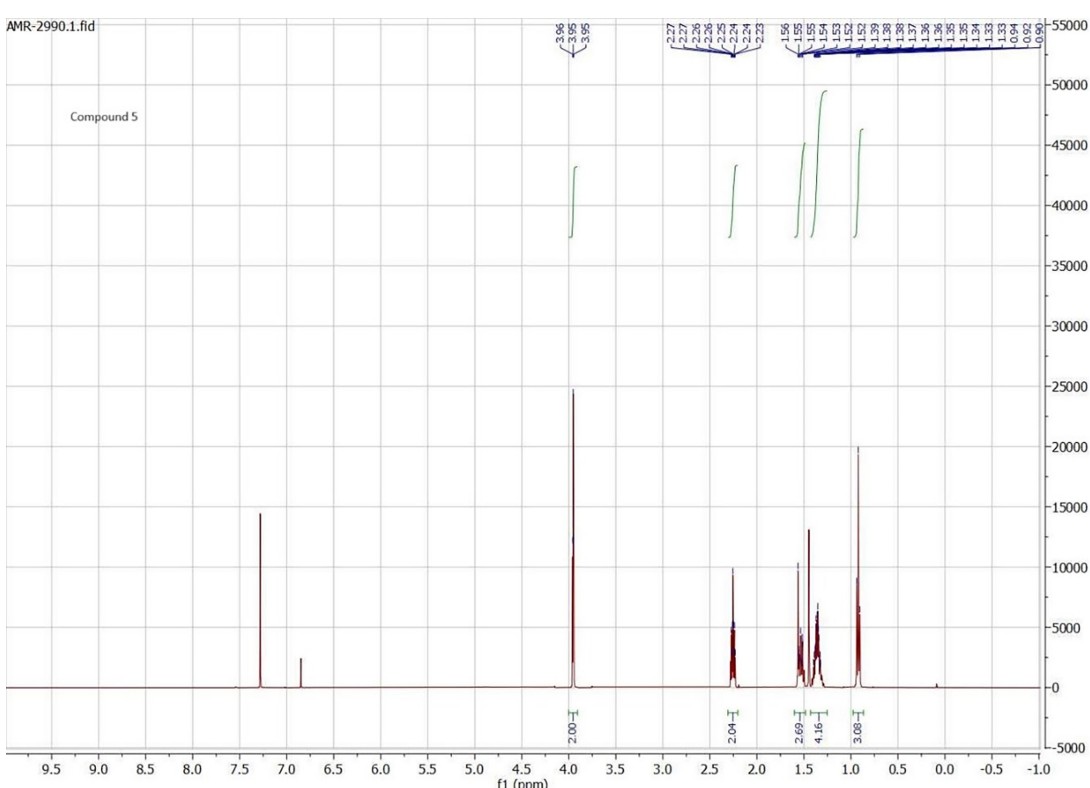

**Appendix 1—figure 9.** ¹H NMR spectrum of compound **5**.

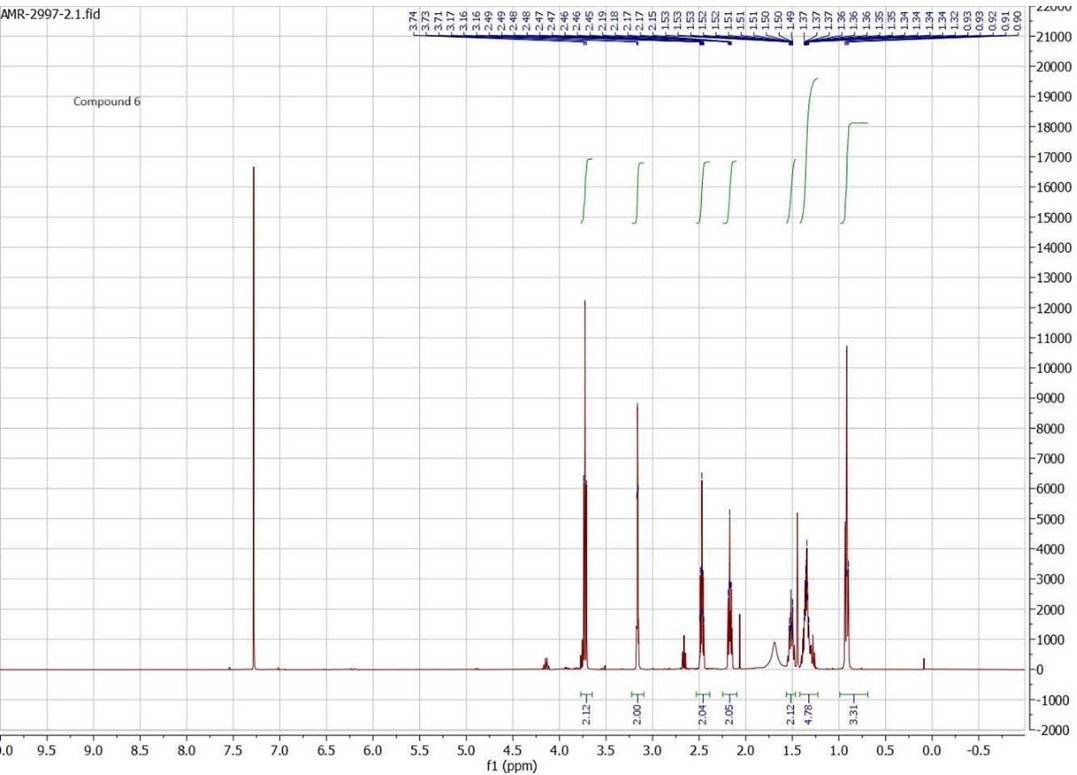

**Appendix 1—figure 10.** [1]H NMR spectrum of compound **6**.

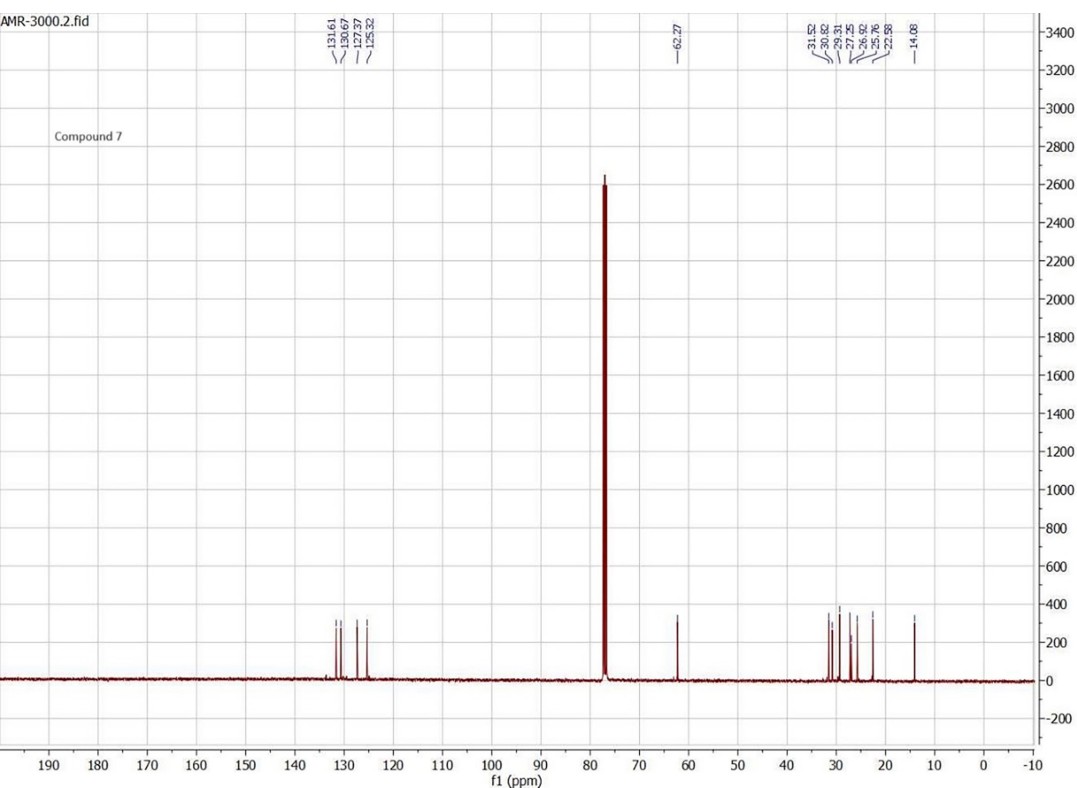

**Appendix 1—figure 11.** [13]C NMR spectrum of compound **7**.

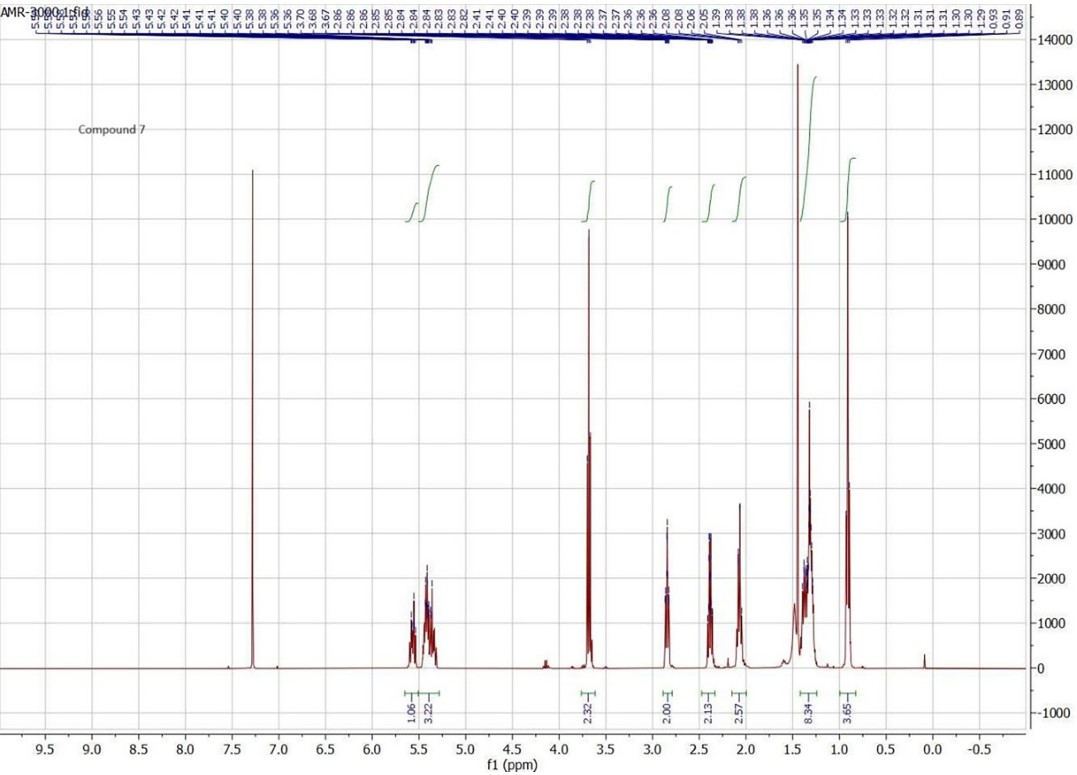

**Appendix 1—figure 12.** $^1$H NMR spectrum of compound **7**.

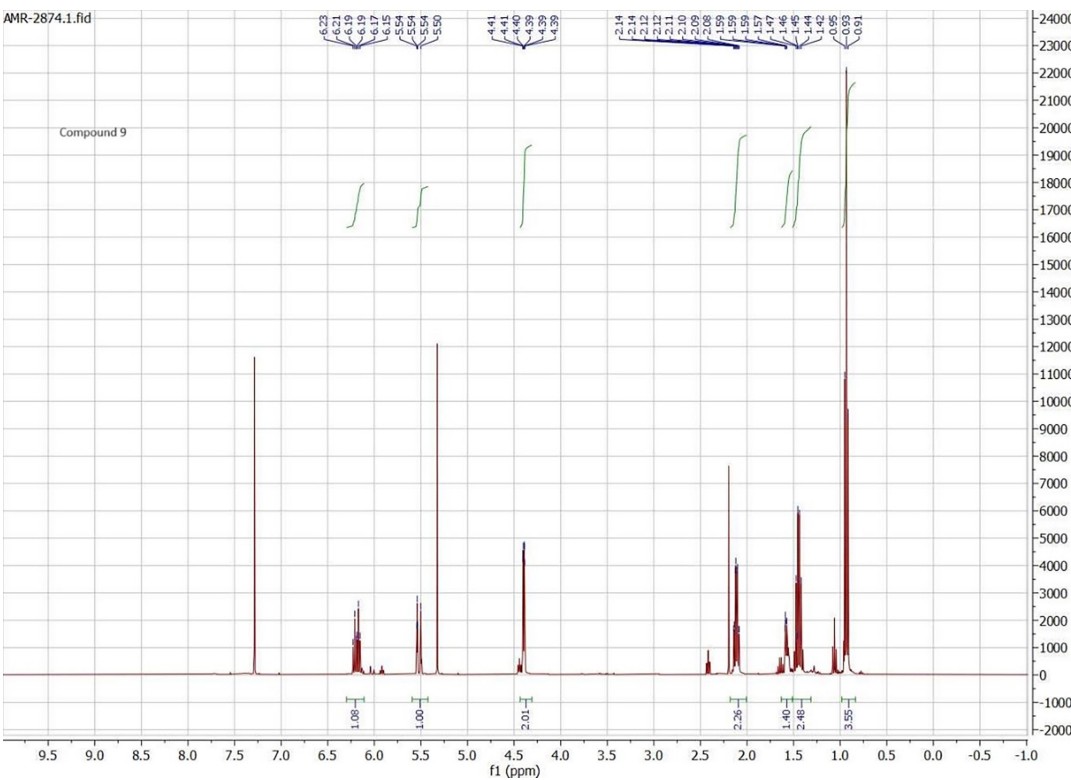

**Appendix 1—figure 13.** $^1$H NMR spectrum of compound **9**.

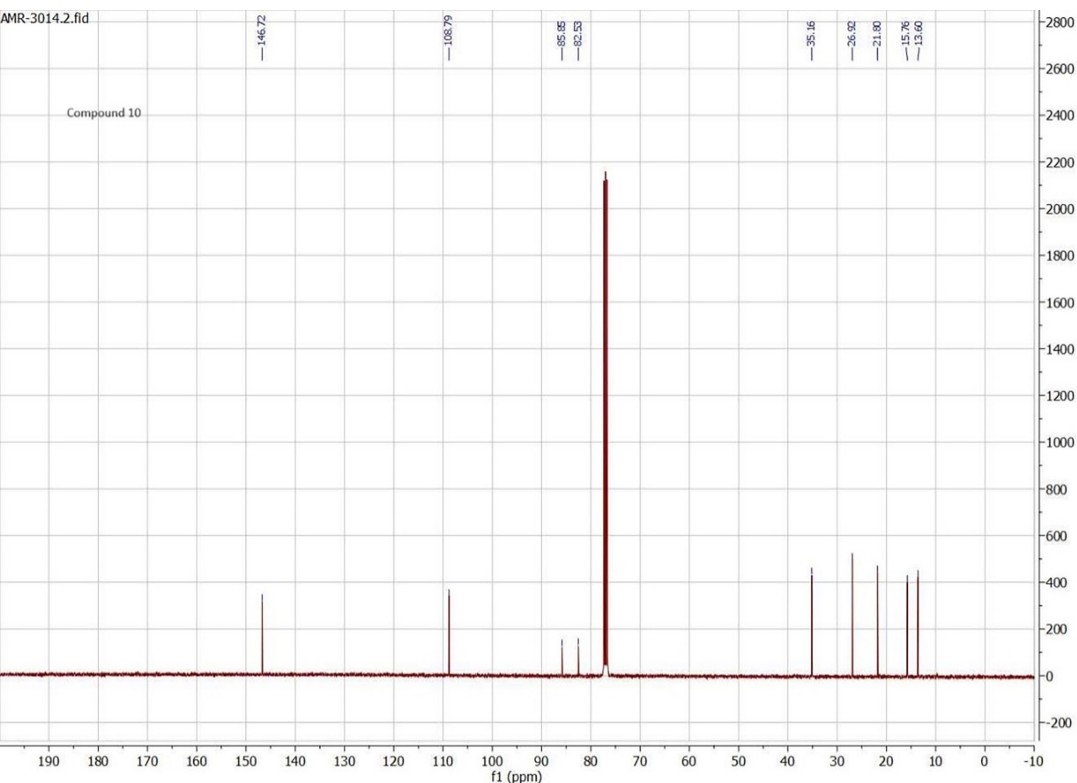

**Appendix 1—figure 14.** $^{13}$C NMR spectrum of compound 10.

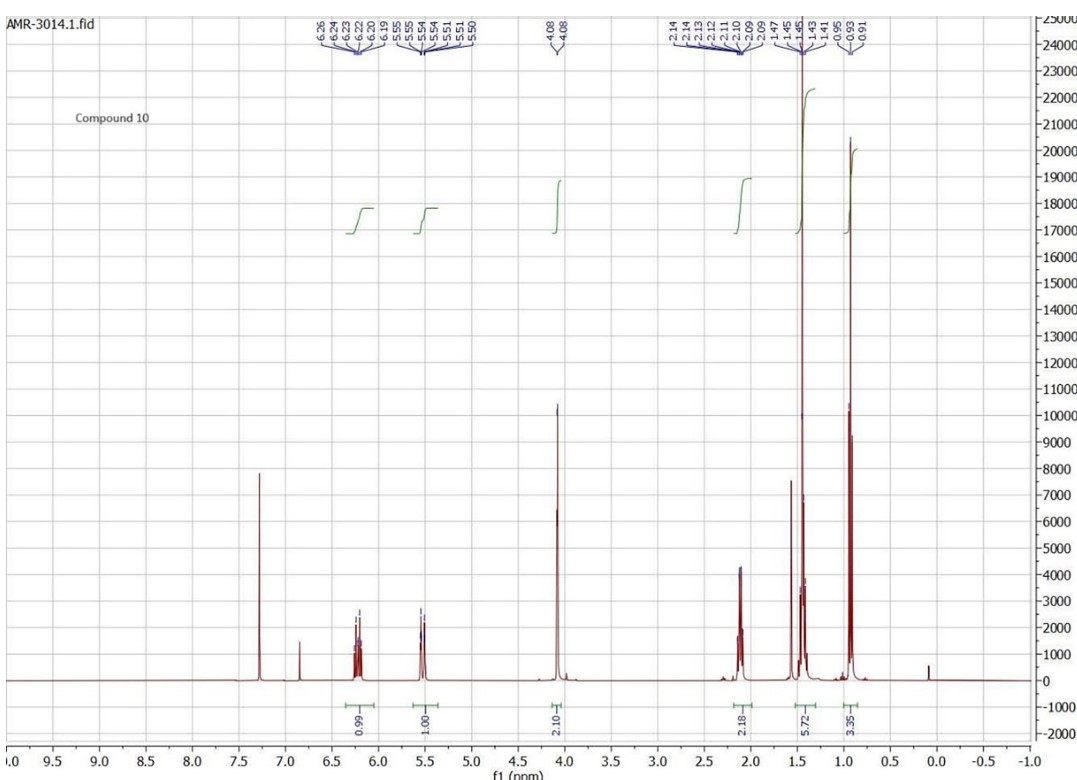

**Appendix 1—figure 15.** $^1$H NMR spectrum of compound 10.

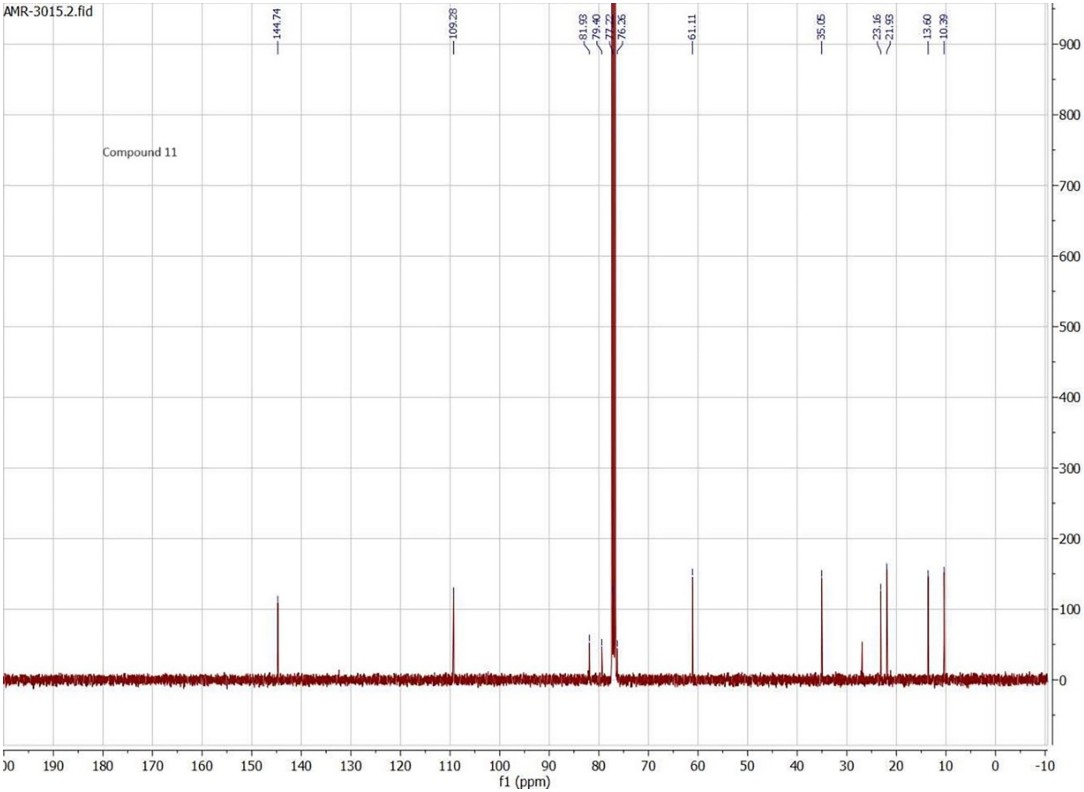

**Appendix 1—figure 16.** $^{13}$C NMR spectrum of compound **11**.

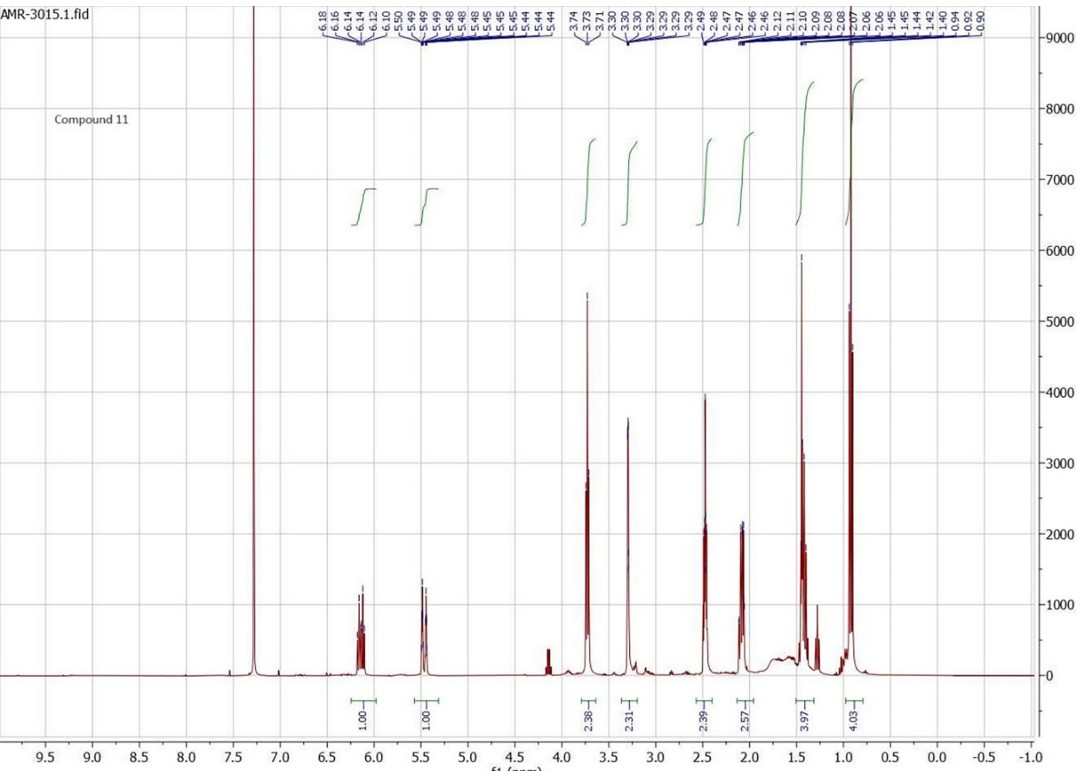

**Appendix 1—figure 17.** $^{1}$H NMR spectrum of compound **11**.

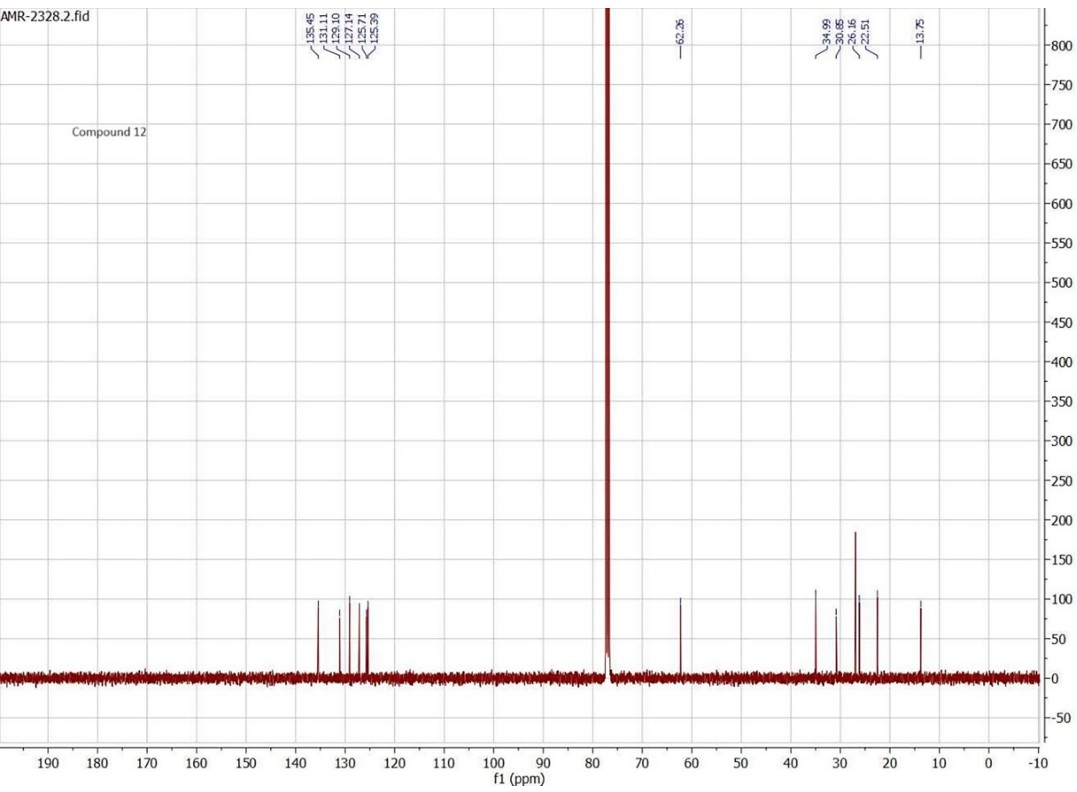

**Appendix 1—figure 18.** $^{13}$C NMR spectrum of compound **12**.

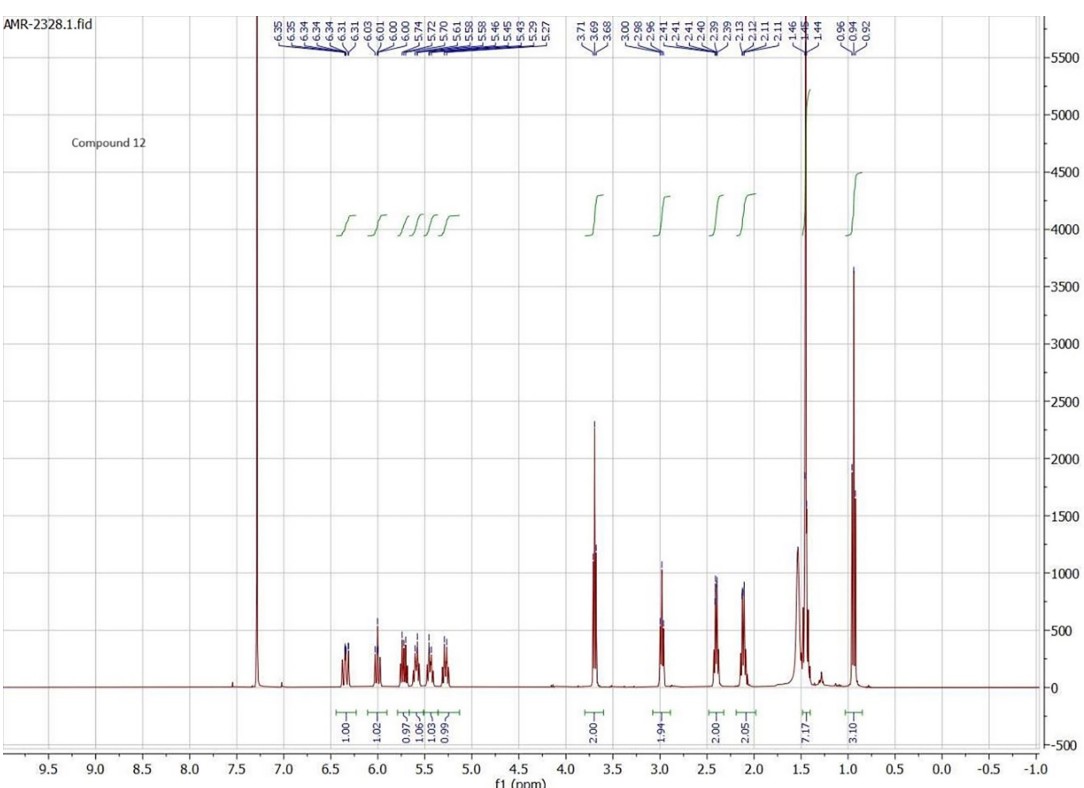

**Appendix 1—figure 19.** $^{1}$H NMR spectrum of compound **12**.

