## [Editor Report · eLife Assessment]

This **important** work by Diallo et al. substantially advances our understanding of the chemosensory system of a non-hymenopteran eusocial insect by identifying the first olfactory receptor for the trail pheromone in termites. The evidence supporting the conclusions that the receptor PsimOR14 is very narrowly tuned for the pheromone neocembrene is **compelling**. The work will be of broad interest to entomologists, chemical ecologists, neuroscientists, and molecular biologists.

---

## [Referee Report · Reviewer #1 (Public review)]

Summary:

In their comprehensive analysis, Diallo et al. deorphanise the first olfactory receptor of a non-hymenopteran eusocial insect - a termite and identified the well established trail pheromone neocembrene as the receptor's best ligand. By using a large set of odorants the authors convincingly show that, as expected for a pheromone receptor, PsimOR14 is very narrowly tuned. While the authors first make use of an ectopic expression system, the empty neuron of *Drosophila melanogaster*, to characterise the receptor's responses, they next perform single sensillum recordings with different sensilla types on the termite antenna. By that they are able to identify a sensillum which houses three neurons, of which the B neuron exhibits the narrow responses described for PsimOR14. Hence the authors do not only identify the first pheromone receptor in a termite but can even localise its expression on the antenna. The authors in addition perform a structural analysis to explain the binding properties of the receptor and its major and minor ligands (as this is beyond my expertise, I cannot judge this part of the manuscript). Finally, they compare expression patterns of ORs in different castes and find that PsimOR14 is more strongly expressed in worker than in soldier termites, which corresponds well with stronger antennal responses in the worker caste.

Strengths:

The manuscript is well written and a pleasure to read.

Weaknesses:

Whenever it comes to the deorphanization of a receptor and its potential role in behaviour (in the case of the manuscript it would be trail following of the termite) one thinks immediately of knocking out the receptor to check whether it is necessary for the behaviour. However, I definitely do not want to ask for this (especially as the establishment of CRISPR Cas-9 in eusocial insects usually turns out to be a nightmare). I also do not know either, whether knock downs via RNAi have been established in termites, but maybe the authors could consider some speculation on this in the discussion.

Comments on revisions:

I appreciate how the authors have replied to my comments and I have the feeling that also the other reviewers' comments have been dealt with carefully. I therefore support the acceptance of this very nice and interesting manuscript.

---

## [Referee Report · Reviewer #2 (Public review)]

Summary:

In this manuscript, the authors performed the functional analysis of odorant receptors (ORs) of the termite Prorhinotermes simplex to identify the receptor of trail-following pheromone. The authors performed single-sensillum recording (SSR) using the transgenic Drosophila flies expressing a candidate of the pheromone receptor and revealed that PsimOR14 strongly responds to neocembrene, the major component of the pheromone. Also, the authors found that one sensillum type (S I) detects neocembrene and also performed SSR for S I in the wild termite workers. Furthermore, the authors revealed the gene, transcript, and protein structures of PsimOR14, predict the 3D model and ligand docking of PsimOR14, and demonstrated that PsimOR14 is higher expressed in workers than soldiers using RNA-seq for heads of workers and soldiers of P. simplex and that EAG response to neocembrene is higher in workers than soldiers. I considered that this study will contribute to further understanding of the molecular and evolutionary mechanisms of chemoreception system in termites.

Strength:

The manuscript is well written. As far as I know, this study is the first study that identified a pheromone receptor in termites. The authors not only present a methodology for analyzing the function of termite pheromone receptors but also provide important insights in terms of the evolution of ligand selectivity of termite pheromone receptors.

Weakness:

This revised manuscript appears to me to have no major weaknesses.

---

## [Referee Report · Reviewer #3 (Public review)]

Summary:

Chemical communication is essential for the organization of eusocial insect societies. It is used in various important contexts, such as foraging and recruiting colony members to food sources. While such pheromones have been chemically identified and their function demonstrated in bioassays, little is known about their perception. Excellent candidates are the odorant receptors that have been shown to be involved in pheromone perception in other insects including ants and bees but not termites. The authors investigated the function of the odorant receptor PsimOR14, which was one of four target odorant receptors based on gene sequences and phylogenetic analyses. They used the Drosophila empty neuron system to demonstrate that the receptor was narrowly tuned to the trail pheromone neocembrene. Similar responses to the odor panel and neocembrene in antennal recordings suggested that one specific antennal sensillum expresses PsimOR14. Additional protein modeling approaches characterized the properties of the ligand binding pocket in the receptor. Finally, PsimOR14 transcripts were found to be significantly higher in worker antennae compared to soldier antennae, which corresponds to the worker's higher sensitivity to neocembrene.

Strengths:

The study presents an excellent characterization of a trail pheromone receptor in a termite species. The integration of receptor phylogeny, receptor functional characterization, antennal sensilla responses, receptor structure modeling, and transcriptomic analysis is especially powerful. All parts build on each other and are well supported with a good sample size. (I cannot comment on protein modeling and docking due to a lack of expertise in this area)

Weaknesses:

None.

---

## [Author Response]

The following is the authors’ response to the original reviews.

**Public Reviews:**

**Reviewer #1 (Public review):**
Summary:In their comprehensive analysis Diallo et al. deorphanise the first olfactory receptor of a nonhymenopteran eusocial insect - a termite and identified the well-established trail pheromone neocembrene as the receptor's best ligand. By using a large set of odorants the authors convincingly show that, as expected for a pheromone receptor, PsimOR14 is very narrowly tuned. While the authors first make use of an ectopic expression system, the empty neuron of *Drosophila melanogaster*, to characterise the receptor's responses, they next perform single sensillum recordings with different sensilla types on the termite antenna. By that, they are able to identify a sensillum that houses three neurons, of which the B neuron exhibits the narrow responses described for PsimOR14. Hence the authors do not only identify the first pheromone receptor in a termite but can even localize its expression on the antenna. The authors in addition perform a structural analysis to explain the binding properties of the receptor and its major and minor ligands (as this is beyond my expertise, I cannot judge this part of the manuscript). Finally, they compare expression patterns of ORs in different castes and find that PsimOR14 is more strongly expressed in workers than in soldier termites, which corresponds well with stronger antennal responses in the worker caste.Strengths:The manuscript is well-written and a pleasure to read. The figures are beautiful and clear. I actually had a hard time coming up with suggestions.

We thank the reviewer for the positive comments.

Weaknesses:Whenever it comes to the deorphanization of a receptor and its potential role in behaviour (in the case of the manuscript it would be trail-following of the termite) one thinks immediately of knocking out the receptor to check whether it is necessary for the behaviour. However, I definitely do not want to ask for this (especially as the establishment of CRISPR Cas-9 in eusocial insects usually turns out to be a nightmare). I also do not know either, whether knockdowns via RNAi have been established in termites, but maybe the authors could consider some speculation on this in the discussion.

We agree that a functional proof of the PsimOR14 function using reverse genetics would be a valuable addition to the study to firmly establish its role in trail pheromone sensing. Nevertheless, such a functional proof is difficult to obtain. Due to the very slow ontogenetic development inherent to termites (several months from an egg to the worker stage) the CRISPR Cas-9 is not a useful technique for this taxon. By contrast, termites are quite responsive to RNAimediated silencing and RNAi has previously been used for the silencing of the ORCo co-receptor in termites resulting in impairment of the trail-following behavior (DOI: 10.1093/jee/toaa248). Likewise, our previous experiments showed a decreased ORCo transcript abundance, lower sensitivity to neocembrene and reduced neocembrene trail following upon dsPsimORCo administration to *P. simplex* workers, while we did not succeed in reducing the transcript abundance of PsimOR14 upon dsPsimOR14 injection. We do not report these negative results in the present manuscript so as not to dilute the main message. In parallel, we are currently developing an alternative way of dsRNA delivery using nanoparticle coating, which may improve the RNAi experiments with ORs in termites.

**Reviewer #2 (Public review):**
Summary:In this manuscript, the authors performed the functional analysis of odorant receptors (ORs) of the termite Prorhinotermes simplex to identify the receptor of trail-following pheromone. The authors performed single-sensillum recording (SSR) using the transgenic *Drosophila* flies expressing a candidate of the pheromone receptor and revealed that PsimOR14 strongly responds to neocembrene, the major component of the pheromone. Also, the authors found that one sensillum type (S I) detects neocembrene and also performed SSR for S I in wild termite workers. Furthermore, the authors revealed the gene, transcript, and protein structures of PsimOR14, predicted the 3D model and ligand docking of PsimOR14, and demonstrated that PsimOR14 is higher expressed in workers than soldiers using RNA-seq for heads of workers and soldiers of *P. simplex* and that EAG response to neocembrene is higher in workers than soldiers. I consider that this study will contribute to further understanding of the molecular and evolutionary mechanisms of the chemoreception system in termites.Strength:The manuscript is well written. As far as I know, this study is the first study that identified a pheromone receptor in termites. The authors not only present a methodology for analyzing the function of termite pheromone receptors but also provide important insights in terms of the evolution of ligand selectivity of termite pheromone receptors.

We thank the reviewer for the overall positive evaluation of the manuscript.

Weakness:As you can see in the "Recommendations to the Authors" section below, there are several things in this paper that are not fully explained about experimental methods. Except for this point, this paper appears to me to have no major weaknesses.

We address point by point the specific comments listed in the Recommendation to the authors chapter below.

**Reviewer #3 (Public review):**
Summary:Chemical communication is essential for the organization of eusocial insect societies. It is used in various important contexts, such as foraging and recruiting colony members to food sources. While such pheromones have been chemically identified and their function demonstrated in bioassays, little is known about their perception. Excellent candidates are the odorant receptors that have been shown to be involved in pheromone perception in other insects including ants and bees but not termites. The authors investigated the function of the odorant receptor PsimOR14, which was one of four target odorant receptors based on gene sequences and phylogenetic analyses. They used the *Drosophila* empty neuron system to demonstrate that the receptor was narrowly tuned to the trail pheromone neocembrene. Similar responses to the odor panel and neocembrene in antennal recordings suggested that one specific antennal sensillum expresses PsimOR14. Additional protein modeling approaches characterized the properties of the ligand binding pocket in the receptor. Finally, PsimOR14 transcripts were found to be significantly higher in worker antennae compared to soldier antennae, which corresponds to the worker's higher sensitivity to neocembrene.Strengths:The study presents an excellent characterization of a trail pheromone receptor in a termite species. The integration of receptor phylogeny, receptor functional characterization, antennal sensilla responses, receptor structure modeling, and transcriptomic analysis is especially powerful. All parts build on each other and are well supported with a good sample size.

We thank the reviewer for these positive comments.

Weaknesses:The manuscript would benefit from a more detailed explanation of the research advances this work provides. Stating that this is the first deorphanization of an odorant receptor in a clade is insufficient. The introduction primarily reviews termite chemical communication and deorphanization of olfactory receptors previously performed. Although this is essential background, it lacks a good integration into explaining what problem the current study solves.

We understand the comment about the lack of an intelligible cue to highlight the motivation and importance of the present study. In the current version of the manuscript the introduction has been reworked. As suggested by Reviewer 3 in the Recommendations section below, the introduction now integrates some parts of the original discussion, especially the part discussing the OR evolution and emergence of eusociality in hymenopteran social insects and in termites, while underscoring the need of data from termites to compare the commonalities and idiosyncrasies in neurophysiological (pre)adaptations potentially linked with the independent eusociality evolution in the two main social insect clades.

Selecting target ORs for deorphanization is an essential step in the approach. Unfortunately, the process of choosing these ORs has not been described. Were the authors just lucky that they found the correct OR out of the 50, or was there a specific selection process that increased the probability of success?

Indeed, we were extremely lucky. Our strategy was to first select a modest set of ORs to confirm the feasibility of the Empty Neuron *Drosophila* system and newly established SSR setup, while taking advantage of having a set of termite pheromones, including those previously identified in the *P. simplex* model, some of them *de novo* synthesized for this project. The selection criteria for the first set of four receptors were (i) to have full-length ORF and at least 6 unambiguously predicted transmembrane regions, and (ii) to be represented on different branches (subbranches) of the phylogenetic tree. Then it was a matter of a good luck to hit the PsimOR14 selectively responding to the genuine *P. simplex* trail-following pheromone main component. In the revised version, we state these selection criteria in the results section (Phylogenetic reconstruction and candidate OR selection).

The deorphanization attempts of additional *P. simplex* ORs are currently running.

The authors assigned antennal sensilla into five categories. Unfortunately, they did not support their categories well. It is not clear how they were able to differentiate SI and SII in their antennal recordings.

We agree that the classification of multiporous sensilla into five categories lacks robust discrimination cues. The identification of the neocembrene-responding sensillum was initially carried out by SSR measurements on individual olfactory sensilla of *P. simplex* workers one-by-one and the topology of each tested sensillum was recorded on optical microscope photographs taken during the SSR experiment. Subsequently, the SEM and HR-SEM were performed in which we localized the neocembrene sensillum and tried to find distinguishing characters. We admit that these are not robust. Therefore, in the revised version of the manuscript we decided to abandon the attempt of sensilla classification and only report the observations about the specific sensillum in which we consistently recorded the response to neocembrene (and geranylgeraniol). The modifications affect Fig. 4, its legend and the corresponding part of the results section (Identification of *P. simplex* olfactory sensillum responding to neocembrene).

The authors used a large odorant panel to determine receptor tuning. The panel included volatile polar compounds and non-volatile non-polar hydrocarbons. Usually, some heat is applied to such non-volatile odorants to increase volatility for receptor testing. It is unclear how it is possible that these non-volatile compounds can reach the tested sensilla without heat application.

The reviewer points at an important methodological error we made while designing the experiments. Indeed, the inclusion of long-chain hydrocarbons into Panel 1 without additional heat applied to the odor cartridges was inappropriate, even though the experiments were performed at 25–26 °C. We carefully considered the best solution to correct the mistake and finally decided to remove all tested ligands beyond C22 from Panel 1, i.e. altogether five compounds. These changes did not affect the remaining Panels 2-4 (containing compounds with sufficient volatility), nor did they affect the message of the manuscript on highly selective response of PsimOR14 to neocembrene (and geranylgeryniol). In consequence, Figures 2, 3 and 5 were updated, along with the supplementary tables containing the raw data on SSR measurements. In addition, the tuning curve for PsimOR14 was re-built and receptor lifetime sparseness value re-calculated (without any important change). We also exchanged squalene for limonene in the docking and molecular dynamics analysis and made new calculations.

**Recommendations for the authors:**

**Reviewer #1 (Recommendations for the authors):**
(1) L 208: "than" instead of "that"

Corrected.

(2) L 527+527 strange squares (•) before dimensions

Apparently an error upon file conversion, corrected.

(3) L553 "reconstructing" instead of "reconstruct"

Corrected.

(4) Two references Chahda et al. and Chang et al. appear too late in the alphabet.

Corrected. Thank you for spotting this mistake. Due to our mistake the author list was ordered according to the alphabet in Czech language, which ranks CH after H.

**Reviewer #2 (Recommendations for the authors):**
(1) L148: Why did the authors select only four ORs (PsimOR9, 14, 30, and 31) though there are 50 ORs in *P. simplex*? I would like you to explain why you chose them.

Our strategy was to first select a modest set of ORs to confirm the feasibility of the Empty Neuron *Drosophila* system and newly established SSR setup, while taking advantage of having a set of termite pheromones, including those previously identified in the *P. simplex* model, some of them *de novo* synthesized for this project. Then, it was a matter of a good luck to hit the PsimOR14 selectively responding to the genuine *P. simplex* trail-following pheromone main component, while the deorphanization attempts of a set of additional *P. simplex* ORs is currently running. In the revised version of the manuscript, we state the selection criteria for the four ORs studied in the Results section (Phylogenetic reconstruction and candidate OR selection).

(2) L149: Where is Figure 1A? Does this mean Figure 1?

Thank you for spotting this mistake. Fig. 1 is now properly labelled as Fig. 1A and 1B in the figure itself and in the legend. Also the text now either refers to either 1A or 1B.

(3) Figure 1: The authors also showed the transcription abundance of all 50 ORs of *P. simplex* in the right bottom of Figure 1, but there is no explanation about it in the main text.

The heatmap reporting the transcript abundances is now labelled as Fig. 1B and is referred to in the discussion section (in the original manuscript it was referred to on the same place as Fig. 1).

(4) L260-265: The authors confirmed higher expression of PsimOR14 in workers than soldiers by using RNA-seq data and stronger EAG responses of PsimOR14 to neocembrene in workers than soldiers, but I think that confirming the expression levels of PsimOR14 in workers and soldiers by RT-qPCR would strengthen the authors' argument (it is optional).

qPCR validation is a suitable complement to read count comparison of RNA Seq data, especially when the data comes from one-sample transcriptomes and/or low coverage sequencing. Yet, our RNA Seq analysis is based on sequencing of three independent biological replicates per phenotype (worker heads vs. soldier heads) with ~20 millions of reads per sample. Thus, the resulting differential gene expression analysis is a sufficient and powerful technique in terms of detection limit and dynamic range.

We admit that the replicate numbers and origin of the RNA seq data should be better specified since the Methods section only referred to the GenBank accession numbers in the original manuscript. Therefore, we added more information in the Methods section (Bioinformatics) and make clear in the Methods that this data comes from our previous research and related bioproject.

(5) L491: I think that "The synthetic processes of these fatty alcohols are ..." is better.

We replaced the sentence with “The *de novo* organic synthesis of these fatty alcohols is described …”

(6) L525 and 527: There are white squares between the number and the unit. Perhaps some characters have been garbled.

Apparently an error upon file conversion, corrected.

(7) L795: ORCo?

Corrected.

(8) L829-830 & Figure 4: Where is Figure 4D?

Thank you for spotting this mistake from the older version of Figure 4. The SSR traces referred to in the legend are in fact a part of Figure 5. Moreover, Figure 4 is now reworked based on the comments by Reviewer 3.

(9) L860-864: Why did the authors select the result of edgeR for the volcano plot in Figure 7 although the authors use both DESeq2 and edgeR? An explanation would be needed.

Both algorithms, DESeq2 and EdgeR, are routinely used for differential gene expression analysis. Since they differ in read count normalization method and statistical testing we decided to use both of them independently in order to reduce false positives. Because the resulting fold changes were practically identical in both algorithms (results for both analyses are listed in Supplementary table S15), we only reported in Fig. 7 the outputs for edgeR to avoid redundancies. We added in the Results section the information that both techniques listed PsimOR14 among the most upregulated in workers.

**Reviewer #3 (Recommendations for the authors):**
The discussion contains many descriptions that would fit better into the introduction, where they could be used to hint at the study's importance (e.g., 292-311, 381-412). The remaining parts often lack a detailed discussion of the results that integrates details from other insect studies. Although references were provided, no details were usually outlined. It would be helpful to see a stronger emphasis on what we learn from this study.

Along with rewriting the introduction, we also modified the discussion. As suggested, the lines 292-311 were rewritten and placed in the introduction. By contrast, we preferred to keep the two paragraphs 381-412 in the discussion, since both of them outline the potential future interesting targets of research on termite ORs.

As suggested, the discussion has been enriched and now includes comparative examples and relevant references about the broad/narrow selectivity of insect ORs, about the expected breadth of tuning of pheromone receptors vs. ORs detecting environmental cues, about the potential role of additional neurons housed in the neocembrene-detecting sensillum of *P. simplex* workers, etc. From both introduction and discussion the redundant details on the chemistry of termite communication have been removed.

This includes explanations of the advantages of the specific methodologies the authors used and how they helped solve the manuscript's problem. What does the phylogeny solve? Was it used to select the ORs tested? It would be helpful to discuss what the phylogeny shows in comparison to other well-studied OR phylogenies, like those from the social Hymenoptera.

We understand the comment. In fact, our motivation to include the phylogenetic tree of termite ORs was essentially to demonstrate (i) the orthologous nature of OR diversity with few expansions on low taxonomic levels, and (ii) to demonstrate graphically the relationship among the four selected sequences. We do not attempt here for a comprehensive phylogenetic analysis, because it would be redundant given that we recently published a large OR phylogeny which includes all sequences used in the present manuscript and analysed them in the proper context of related (cockroaches) and unrelated insect taxa (Johny et al., 2023). This paper also discusses the termite phylogenetic pattern with those observed in other Insecta. This paper is repeatedly cited on appropriate places of the present manuscript and its main observations are provided in the Introduction section. Therefore, we feel that thorough discussion on termite phylogeny would be redundant in the present paper.

The authors categorized the sensilla types. Potential problems in the categorization aside, it would be helpful to know if it is expected that you have sensilla specialized in perceiving one specific pheromone. What is known about sensilla in other insects?

We understand. In the discussion of the revised version, we develop more about the features typical/expected for a pheromone receptor and the sensillum housing this receptor together with two other olfactory sensory neurons, including examples from other insects.

As the manuscript currently stands, specialist readers with their respective background knowledge would find this study very interesting. In contrast, the general reader would probably fail to appreciate the importance of the results.

We hope that the re-organized and simplified introduction may now be more intelligible even for non-specialist readers.

(1) L35: Should "workers" be replaced with "worker antennae"?

Corrected.

(2) L62: Should "conservativeness" be replaced by "conservation"?

Replaced with “parsimony”.

(3) L129: How and why did the authors choose four candidate ORs? I could not find any information about this in the manuscript. I wondered why they did not pick the more highly expressed PsimOr20 and 26 (Figure 7).

As already replied above in the Weaknesses section, we selected for the first deorphanization attempts only a modest set of four ORs, while an additional set is currently being tested. We also explained above the inclusion criteria, i.e. (i) full-length ORF and at least 6 unambiguously predicted transmembrane regions, and (ii) presence on different branches (subbranches) of the OR phylogeny. For these reasons, we did not primarily consider the expression patterns of different ORs. As for Fig. 7, it shows differential expression between soldiers and workers, which was not the primary guideline either and the data was obtained only after having the ORs tested by SSR. Yet, even though we had data on *P. simplex* ORs expression (Fig. 1B), we did not presume that pheromone receptors should be among the most expressed ORs, given the richness of chemical cues detected by worker termites and unlike, e.g., male moths, where ORs for sex pheromones are intuitively highly expressed.

The strategy of OR selection is specified in the results section of the revised manuscript under “Phylogenetic reconstruction and candidate OR selection”.

(4) 198 to 200: SI, II, and III look very similar. Additional measurements rather than qualitative descriptions are required to consider them distinct sensilla. The bending of SIII could be an artifact of preparation. I do not see how the authors could distinguish between SI and SII under the optical microscope for recordings. A detailed explanation is required.

As we responded above in “Weaknesses” chapter, we admit that the sensilla classification is not intelligible. Therefore, we decided in the revised version to abandon the classification of sensilla types and only focus on the observations made on the neocembreneresponding sensillum. To recognize the specific sensillum, we used its topology on the last antennal segment. Because termite antennae are not densely populated with sensilla, it is relatively easy to distinguish individual sensilla based on their topology on the antenna, both in optical microscope and SEM photographs. The modifications affect Fig. 4, its legend and the corresponding part of the results section (Identification of *P. simplex* olfactory sensillum responding to neocembrene).

(5) 208: "Than" instead of "that"

Corrected.

(6) 280: I suggest replacing "demand" with "capabilities"

Corrected.

(7) 312: Why "nevertheless? It sounds as if the authors suggest that there is evidence that ORs are not important for communication. This should be reworded.

We removed “Nevertheless” from the beginning of the sentence.

(8) 321 to 323: This sentence sounds as if something is missing. I suggest rewriting it.

This sentence simply says that empty neuron *Drosophila* is a good tool for termite OR deorphanization and that termite ORs work well *Drosophila* ORCo. We reworded the sentence.

(9) 323: I suggest starting a new paragraph.

Corrected.

(10) 421: How many colonies were used for each of the analyses?

The data for this manuscript were collected from three different colonies collected in Cuba. We now describe in the Materials and Methods section which analyses were conducted with each of the colonies.

(11) 430: Did the termites originate from one or multiple colonies and did the authors sample from the Florida and Cuba population?

The data for this manuscript were collected from three different colonies collected in Cuba. We now describe in the Materials and Methods section which analyses were conducted with each of the colonies.

(12) 501: How was the termite antenna fixated? The authors refer to the *Drosophila* methods, but given the large antennal differences between these species, more specific information would be helpful.

Understood. We added the following information into the Methods section under “Electrophysiology”: “The grounding electrode was carefully inserted into the clypeus and the antenna was fixed on a microscope slide using a glass electrode. To avoid the antennal movement, the microscope slide was covered with double-sided tape and the three distal antennal segments were attached to the slide.”

(13)509: I want to confirm that the authors indicate that the outlet of the glass tube with the airstream and odorant is 4 cm away from the *Drosophila* or termite antenna. The distance seems to be very large.

Thank you for spotting this obvious mistake. The 4 cm distance applies for the distance between the opening for Pasteur pipette insertion into the delivery tube, the outlet itself is situated approx. 1 cm from the antenna. This information is now corrected.

(14) 510/527: It looks like all odor panels were equally applied onto the filter paper despite the difference in solvent (hexane and paraffin oil). How was the solvent difference addressed?

In our study we combine two types of odorant panels. First, we test on all four studied receptors a panel containing several compounds relevant for termite chemical communication including the C12 unsaturated alcohols, the diterpene neocembrene, the sesquiterpene (3R,6E)-nerolidol and other compounds. These compounds are stored in the laboratory as hexane solutions to prevent the oxidation/polymerization and it is not advisable to transfer them to another solvent. In the second step we used three additional panels of frequently occurring insect semiochemicals, which are stored as paraffin oil solutions, so as to address the breadth of PsimOR14 tuning. We are aware that the evaporation dynamics differ between the two solvents but we did not have any suitable option how to solve this problem. We believe that the use of the two solvents does not compromise the general message on the receptor specificity. For each panel, the corresponding solvent is used as a control. Similarly, the use of two different solvents for SSR can be encountered in other studies, e.g. 10.1016/j.celrep.2015.07.031.

(15) 518: delta spikes/sec works for all tables except for the wild type in Table S5. I could not figure out how the authors get to delta spikes/sec in that table.

Thank you for your sharp eye. Due to our mistake, the values of Δ spikes per second reported in Table S5 for W1118 were erroneously calculated using the formula for 0.5 sec stimulation instead of 1 sec. We corrected this mistake which does not impact the results interpretation in Table S5 and Fig. 2.

522: Did the workers and soldiers originate from different colonies or different populations?

We now clearly describe in the Material and Methods section the origin of termites for different experiments. EAG measurements were made using individuals (workers, soldiers) from one Cuban colony.

(16) Figure 6C/D: I suggest matching colors between the two figures. For example, instead of using an orange circle in C and a green coloration of the intracellular flap in D, I recommend using blue, which is not used for something else. In addition, the binding pocket could be separated better from anything else in a different color.

We agree that the color match for the intracellular flap was missing. This figure is now reworked and the colors should have a better match and the binding region is better delineated.

(17) Figure 7/Table S15: It is unclear where the transcriptome data originate and what they are based on. Are these antennal transcriptomes or head transcriptomes? Do these data come from previous data sets or data generated in this study? Figure 7 refers to heads, Table S15 to workers and soldiers, and the methods only refer to antennal extractions. This should be clarified in the text, the figure, and the table.

We admit that the replicate numbers and origin of the RNA seq data should be better specified and that the information that the RNASeq originated from samples of heads+antennae of workers and soldiers should be provided at appropriate places. Therefore, we added more information on replicates and origin of the data in the Methods section (Bioinformatics) and make clear that this data comes from our previous research and refer to the corresponding bioproject. Likewise, the Figure 7 legend and Table S15 heading have been updated.